# Orbital-scale variability in the contribution of foraminifera and coccolithophores to pelagic carbonate production

Pauline Cornuault[1], Luc Beaufort[2], Heiko Pälike[1,3], Torsten Bickert[1], Karl-Heinz Baumann[1,3], Michal Kucera[1]

[1]University of Bremen, MARUM - Center for Marine Environmental Sciences, Leobener Straße 8, D-28359 Bremen, Germany
[2]Aix Marseille Univ, CNRS, IRD, INRAE, Coll France, CEREGE, Avenue Louis Philibert, F13090 Aix-en-Provence, France
[3]University of Bremen, Geoscience Department, Klagenfurter Straße, PO Box 330440, 28359 Bremen, Germany

*Correspondence to:* Pauline Cornuault (pauline.cornuault@gmail.com)

**Abstract.** Throughout the Cenozoic, calcareous nannoplankton and planktonic foraminifera have been the main producers of pelagic carbonate preserved on the seafloor. While past variability in pelagic carbonate production has been previously studied, relatively little is known about the variability in the relative contribution of the two components. This is important because the responses of the two groups to environmental forcing could be different such that they could amplify or reduce the magnitude of fluctuations in total carbonate production. Here we present new data from the tropical Atlantic that allow us to quantify changes in the relative contribution of the two groups to the total pelagic carbonate burial flux on orbital scales and between different climate states since the Miocene. We find that the composition of the deposited pelagic carbonate remained similar on long time scales, with foraminifera making up about 30% of the deposited carbonate, but varied by up to a factor of two on orbital time scales. We show that the relative contribution of planktonic foraminifera and coccoliths did not correlate with the total pelagic carbonate production, either in the Pliocene, when its dominant cyclicity was in the precession band and in phase with orbital parameters modulations, or in the Miocene, when its predominant cyclicity was in the eccentricity band and in antiphase with orbital parameters modulations. The observed variability in tropical pelagic carbonate productivity between foraminifera and coccolithophores suggests that the two main groups of pelagic calcifiers responded fundamentally differently to orbital forcing and associated oceanographic changes in the tropical ocean, but the resulting changes in their proportions did not drive changes in overall pelagic carbonate deposition on either geological or orbital time scales.

## 1 Introduction

The biomineralisation of carbonate by pelagic calcifiers (planktonic foraminifera and calcareous nannofossils) is a key element of the marine carbon cycle, facilitating the long-term removal of carbon into the sedimentary reservoir (Landschützer et al., 2014). This reservoir interacts with the marine carbon cycle in various ways, such as chemical compensation, where sedimentary carbonate dissolution can compensate for changes in ocean carbonate chemistry, and biological compensation, which refers to changes in the amount of carbonate biomineralisation, removing dissolved carbonate and bicarbonate, thereby lowering oceanic alkalinity (Boudreau et al., 2018). Remarkably, modelling studies show that a change in carbonate production of only 10% on short (ka) to geological time scales would be sufficient to affect the marine carbon cycle. A global decrease in carbonate biomineralisation would lead to a higher alkalinity and, thus, to a higher capacity of the ocean to absorb dissolved $CO_2$ (Boudreau et al., 2018). Both biological and chemical compensation depend not only on the total amount of oceanic biomineralisation, but also on the composition of the deposited carbonate (Si and Rosenthal, 2019). This is because the main components of the pelagic calcite flux to the seafloor, planktonic foraminifera and calcareous nannoplankton, produce skeletons of very different sizes and shapes and thus different sinking behaviour and susceptibility to dissolution. Calcareous nannoplankton is up to two orders of magnitude smaller than foraminifera shells but can sink rapidly as it is often concentrated in faecal pellets (Fischer and Karakaş, 2009; Richardson and Jackson, 2007; Ziveri et al., 2007). The shells of foraminifera are

composed of nanoscale crystal units with a large surface-to-volume ratio, making their shells more susceptible to dissolution (Honjo and Erez, 1978).

While total pelagic calcite production and burial and its changes on geological time scales have been intensively studied locally and globally, the relative contribution of planktonic foraminifera and coccolithophores to the burial flux remains less well constrained. Overall, coccolithophore calcite is considered to be the main component of deep sea pelagic carbonate, but estimates of the contribution of coccoliths to the biogenic calcite exported from the photic zone vary substantially (20-80%), reflecting differences among regions and methods used to quantify the mass of coccolithophore calcite (Baumann et al., 2004; Frenz et al., 2005, 2006; Ramaswamy and Gaye, 2006; Schiebel, 2002), and there is strong evidence that the composition of the pelagic carbonate flux varied on geological time scales (Chiu and Broecker, 2008; Si and Rosenthal, 2019). Next to the processes that may modify the biogenic carbonate flux during sinking through the water column, there are principally three primary factors that can influence the composition of pelagic carbonate leaving the productive zone: the population size of the producers, their mean cell size, and their investment in biomineralisation relative to cell size. Because coccolithophores and planktonic foraminifera follow different lifestyles, have different metabolisms, and have different environmental preferences, they may respond differently to environmental change with respect to each of the three factors, resulting in differences in the pelagic calcite flux composition (Gehlen et al., 2007; Langer, 2008; Si and Rosenthal, 2019). Si and Rosenthal (2019) show a long geological time scale (millions of years) shift towards more foraminifera and proportionally fewer coccoliths towards the Quaternary. They attribute it to long-term decrease in ocean alkalinity supplied by continental weathering, due to decreasing $pCO_2$ throughout this time interval. In our study presented here, we investigate these variations at geological and orbital time scales. To understand to what degree the composition of pelagic carbonate flux varied in the past and to qualify and quantify the changes in the differential contribution of the two main pelagic calcifiers to the total pelagic carbonate production estimated from the carbonate accumulation rate ($CaCO_3$ AR) (Brummer and van Eijden, 1992; Liebrand et al., 2016), we generated new data for Leg 154 ODP Site 927, Ceará Rise. This Site was chosen for its good carbonate preservation during Quaternary interglacials and throughout the Neogene (Curry et al., 1995) and because the site remained in a tropical setting throughout, while at the same time, well-recorded orbital cyclicity of sediment properties allow the development of a tuned age model and quantification of the flux at orbital resolution. We made use of previously analysed samples (Cornuault et al., 2023) and determined the contribution of coccoliths and foraminifera to the total carbonate production using two independent methods to estimate the coccolith fraction. The sampling strategy allowed us to quantify changes in the composition of the sedimentary carbonate at orbital time scale in the Pliocene and Miocene and among in four time intervals known to be potential analogs for today's climate warming conditions: marine isotopic stage (MIS) 5 (88 to 150 ka, Clark and Huybers, 2009; Kopp et al., 2009), MIS 9 (276 to 370 ka, (Past Interglacials Working Group of PAGES, 2016; Voelker et al., 2010)), the Pliocene warm period (PWP, MIS KM5, 3096 to 3307 ka, Ravelo et al., 2004) and the Miocene climatic optimum (MCO, 15.6 to 16 Ma, Foster et al., 2012; Pound et al., 2012; You et al., 2009; Zachos et al., 2008).

## 2 Material and methods

ODP Site 927 on the Ceará Rise in the western tropical Atlantic Ocean (5°27.77'N, 44°28.84'W, 3315 metres below sea level (mbsl)) is located above the modern regional lysocline (4200 mbsl, Cullen and Curry, 1997; Curry et al., 1995; Frenz et al., 2006). However, the depth of the lysocline has varied in the past, and some of the shoaling episodes have resulted in the site being affected by carbonate dissolution. Such episodes are known from the glacial periods of the late Quaternary and are related to the restructuring of the Atlantic Meridional Overturning Circulation (AMOC) and an increased influence of the more corrosive Antarctic bottom waters (Gröger et al., 2003b, a). During the studied intervals of the Pliocene and Miocene, no such events occurred at this site. Indeed, for those two periods, unlike for the Quaternary, there is no relationship between carbonate content and carbonate flux (Cornuault et al., 2023), which could indicate dissolution. In addition, the Pliocene and Miocene

sediments from the studied site are continuously carbonate rich and their biogenic fraction contains virtually no siliceous components (Curry et al., 1995). Finally, the preservation of planktonic foraminifera shells remains good throughout the studied intervals, indicating deposition above the regional lysocline (Curry et al., 1995; Gröger et al., 2003b). The location of this site is far from the high-latitude climate changes and large-scale temperature variations at time scales from thousand years to million years. From this site, we already have a set of samples corresponding to these four time intervals, of which we

already know the carbonate content and have a high resolution tuned age model (Cornuault et al., 2023). The carbonate content analyses on the bulk sediment were done using a LECO CS744 elemental analyser at Bremen University (Cornuault et al., 2023). Thanks to their good state of preservation, their high carbonate content and sampling at orbital resolution, the data allow us to study both planktonic foraminifera and coccoliths in the same samples, preventing a possible bias on the relative changes between the two groups with respect to timing (and potential lags) of their response.

**2.1 Carbonate content of the different size classes and their relative contribution to the total pelagic carbonate production**

To characterise the contribution of the planktonic foraminifera and the coccoliths to the total carbonate, we have used two different and independent approaches. In the first approach, we assumed that most of the foraminifera shells and the coccoliths can be separated by size. At the studied location the carbonate is mainly composed of foraminifera and coccoliths and a very

low portion of the carbonate content is composed of shell fragments; there are virtually no siliceous biogenic components in the sediments (Curry et al., 1995). During the studied time intervals and at this location the coccoliths have a median size around 4 µm diameter and the foraminifera have a median maximum diameter between 110 and 250 µm. Further, we assumed that most of the foraminifera calcite mass occurs in the size fraction above 63 µm and therefore, that the carbonate flux in the >63 µm size fraction represents foraminifera $CaCO_3$ flux (Si and Rosenthal, 2019). Some foraminifera shells and fragments

are inevitably also present in the <63 µm size fraction, however, existing estimates indicate that the contributions of such small shells and fragments to the total foraminifera calcite mass is negligible (Kiss et al., 2021 and references therein).

The bulk sediment samples were disaggregated in tap water in 15 mL centrifuge tubes in a rotating carousel overnight and washed and sieved at 63 µm. The coarse fraction was then dried at 50°C overnight. Next, the dry bulk sediment (DBS) weight and the coarse fraction weight were used to calculate the proportion of the coarse fraction relative to the dry bulk sediment:

Coarse fraction (>63 µm)% = (>63 µm (g) / DBS (g))*100. Then, knowing the bulk carbonate content (Cornuault et al., 2023) and assuming the coarse fraction only consisted of carbonate (foraminifera shells), we calculated the percentage contribution of the <63 µm size fraction to the total carbonate content of the sediment. Following the approach by Si and Rosenthal (2019), we then calculated the contribution of the coccoliths assuming it represents all the carbonate in the <63 µm size fraction. Finally, using the sedimentation rate (SR) and the dry bulk density (DBD), we derived the accumulation rate of both

components as an estimate of their export production (Liebrand et al., 2016).

In the second approach, because the <63 µm size fraction may contain non-coccolith carbonate particles, and to explore the effect of the simplification of using 63 µm as a size threshold, we generated an independent record of coccolith fraction $CaCO_3$ AR, based on automated image analysis of the <32 µm size fraction of the sediment. This threshold is commonly used (Beaufort et al., 2014) because this mesh size still allows effective mechanical size fractionation, whilst assuring that all coccolith-

derived particles are in the smaller fraction and the majority of the foraminifera shells (with the exception of the tiniest fragments) remain in the coarse fraction. To this end, we prepared microscope slides using a quantitative protocol to determine the amount of sediment <32 µm that was deposited on the slide and measured using optical microscopy the amount of carbonate <32 µm it contained. To assure a random and even distribution of the material on the slide, we followed the protocol presented by Beaufort et al. (2014). In this protocol, a tiny quantity of sediment is disaggregated in water in a vial and poured in a receiver

containing a 12 mm x 12 mm coverslip, over a <32 µm mesh metal sieve. Specifically, we used ~ 0.1 $mm^3$ of bulk sediment in 1 mL of low alkaline water in a 15 mL centrifuge tube. The quantity of the deposited sediment was determined by weighing

of the microslide using an ultra-microbalance with nominal precision of 0,1 µg. The resulting slides were analysed at CEREGE, France, using an automated microscope with the SYRACO device that automatically takes optical microscope pictures with polarised light (Beaufort et al., 2014; Beaufort and Dollfus, 2004) and provides an estimation of the weight of calcite per

coccolith analysed and the total mass of calcite contained in the analysed field of view (FOV). The version of SYRACO used in this study is the one described by Beaufort et al. (2022). Knowing the number of FOV analysed and the number of FOV on the whole slide, we can calculate the quantity of carbonate on the whole slide. As we know the quantity of bulk sediment on the slide, we can then calculate the coccolith carbonate content of the small fraction, as the ratio between the carbonate mass estimated by SYRACO and the mass of bulk sediment. We note that the resulting output is not directly comparable to the first

method because of a different size fraction used and because of an entirely different methodology that also depends on the accuracy of the scaling of the SYRACO procedure, but the method should provide a valid alternative estimate of the amount and nature of variation in the flux of coccolith calcite across the studied interval (Figures 2, S2 and S3). With regard to the absolute values of coccolith calcite flux, the estimate from SYRACO should be consistently lower than that of the first method, not only because of the smaller size fraction range used but also because SYRACO only analyses well-preserved, well-focused

specimens that are not part of aggregates.

## 2.2 Spectral analysis

To see whether the CaCO$_3$ AR changes of each size fraction are orbitally driven or not, we performed a Wavelet Transform (WT) spectral analysis (WaveletComp 1.1 package on R, Roesch and Schmidbauer, 2018) using R (4.1.2., R Core Team, 2021) for all parameters for the Pliocene and the Miocene intervals (except for the Quaternary, as these two time intervals are marked

by strong precession-paced dissolution cycles, Harris et al., 1997). Additionally, to observe the actual relationship between the changes in the relative contribution of the two size fractions and the environmental conditions at orbital time scale, we compared our results to an E+T-P (eccentricity plus tilt minus precession) record (Laskar et al., 2004) that reflects the different orbital parameters.

## 3 Results

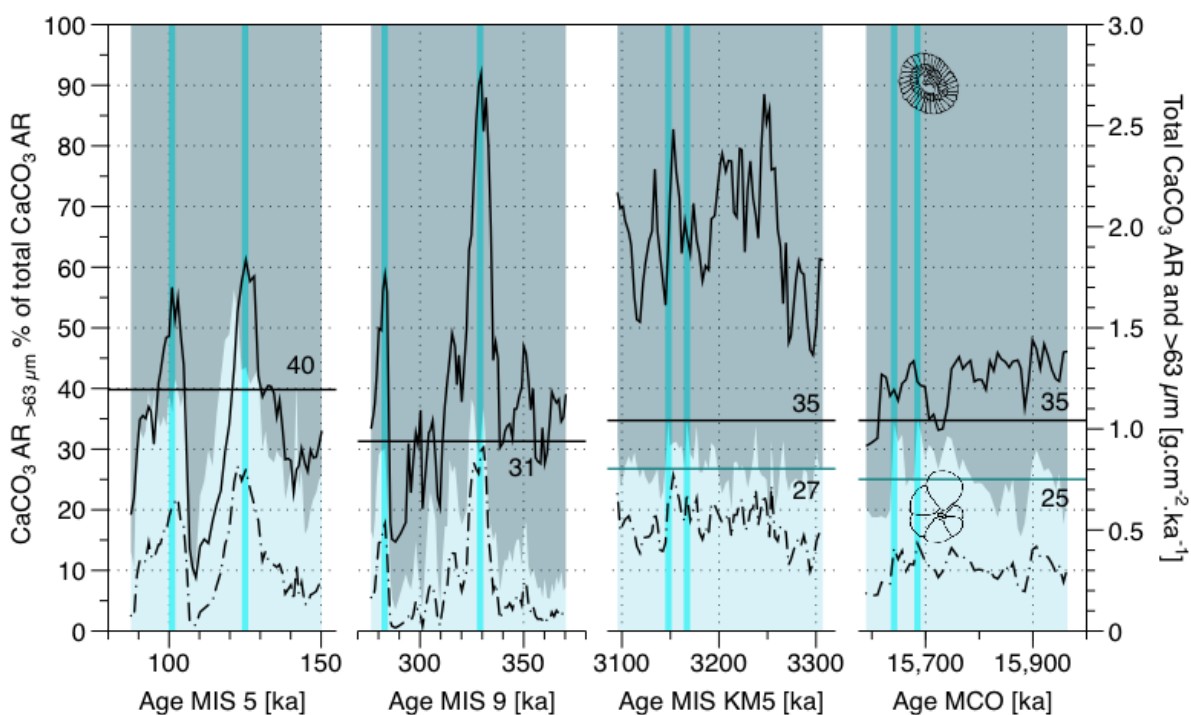


**Figure 1: Variability of the >63 µm carbonate contribution to the total CaCO₃ (background fill) and the CaCO₃ accumulation rate of the total sediment (black solid line) and of the >63 µm size fractions (black dashed line). Representative values (solid black lines with numbers) of the contribution of the coarse fraction to the bulk carbonate accumulation rate were calculated for each of the four time intervals as means of the two highest coarse contribution values (highlighted by vertical blue lines) in each interval. This was done to account for the effect of dissolution on the Quaternary samples. For MIS KM5 and MCO, where none of the samples were affected by dissolution, the overall averages per intervals are shown by blue horizontal lines with numbers.**

The estimated contribution of foraminifera to the sedimentary carbonate based on particle size fractionation varies between 3.5% and 56.4% (Figure 1). The maximum values of the carbonate fraction of foraminifera for the four intervals studied are quite similar, ranging from 31.3% for MIS 9 to 39.8% for MIS 5 (Figure 1). The largest variations occurred on orbital scale in the Quaternary, but a variability on orbital scales was also present during the Pliocene and Miocene (Figure 1) and is also evident in the record of coccolith fraction accumulation obtained by the SYRACO method (Figure S3).

During the Quaternary, the estimated percent contribution of foraminifera to the sedimentary carbonate co-varied and was in phase with the bulk carbonate accumulation rate as well as the foraminifera fraction accumulation rate (Figure 1). In contrast, during the Neogene, the contribution of the coarse fraction (foraminifera) to the CaCO₃ AR bulk and the total CaCO₃ AR bulk do not appear to correlate.

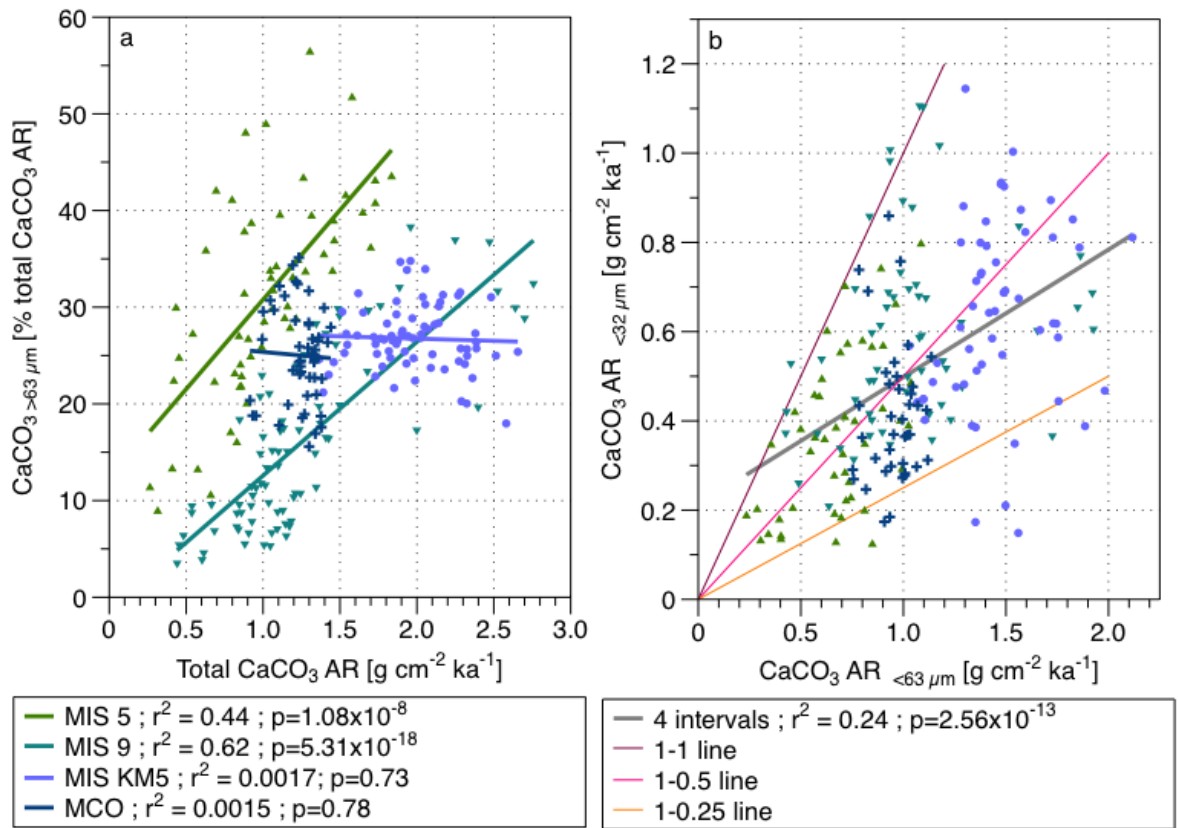

| | |
|---|---|
| — MIS 5 ; $r^2$ = 0.44 ; p=1.08x10$^{-8}$ | — 4 intervals ; $r^2$ = 0.24 ; p=2.56x10$^{-13}$ |
| — MIS 9 ; $r^2$ = 0.62 ; p=5.31x10$^{-18}$ | — 1-1 line |
| — MIS KM5 ; $r^2$ = 0.0017; p=0.73 | — 1-0.5 line |
| — MCO ; $r^2$ = 0.0015 ; p=0.78 | — 1-0.25 line |

**Figure 2: a) Contribution of the >63 µm fraction to the total CaCO₃ AR versus total CaCO₃ AR for the four studied time intervals, with regression lines and b) the coccolith CaCO₃ determined by SYRACO versus the CaCO₃ AR of the <63 µm fraction, with a common regression line (grey) of the four studies intervals and coloured lines showing the difference expressed as percentage underestimation by the SYRACO method MIS 5 values are green triangles, MIS 9 values the teal triangles, MIS KM5 are the purple dots and MCO the dark blue crosses.**

To test how the relative contribution of foraminifera and coccoliths is related to the total CaCO₃ AR, we plotted the contribution of the coarse fraction to the bulk CaCO₃ AR vs the CaCO₃ AR bulk (Figure 2a). For the Quaternary intervals, which were affected by dissolution, we observe a strong positive correlation. For the two Neogene intervals that were not affected by carbonate dissolution, there is no correlation between the coarse fraction contribution to the total CaCO₃ AR and the bulk CaCO₃, indicating that the relative contribution of the two pelagic calcifiers was not driving the total bulk CaCO₃ AR (Figure

2a).

To test if the results may be affected by the way we estimated the percentage of coccolith carbonate in the samples, we plotted the CaCO₃ AR <32 µm estimated using the SYRACO device over the CaCO₃ AR <63 µm estimated using the Si and Rosenthal (2019) approach (Figure 2b). If the values obtained from two different methods reflect the same underlying variability, we would expect them to be correlated with the SYRACO values being systematically lower and showing no difference in the

shape of the relationship among the four studied intervals. Indeed, we observe a strong positive relationship between the CaCO₃ AR estimation of the small fraction of the two methods ($r^2 = 0.24$), as well as a systematically lower values given by the SYRACO method, as expected (Figure 2b).

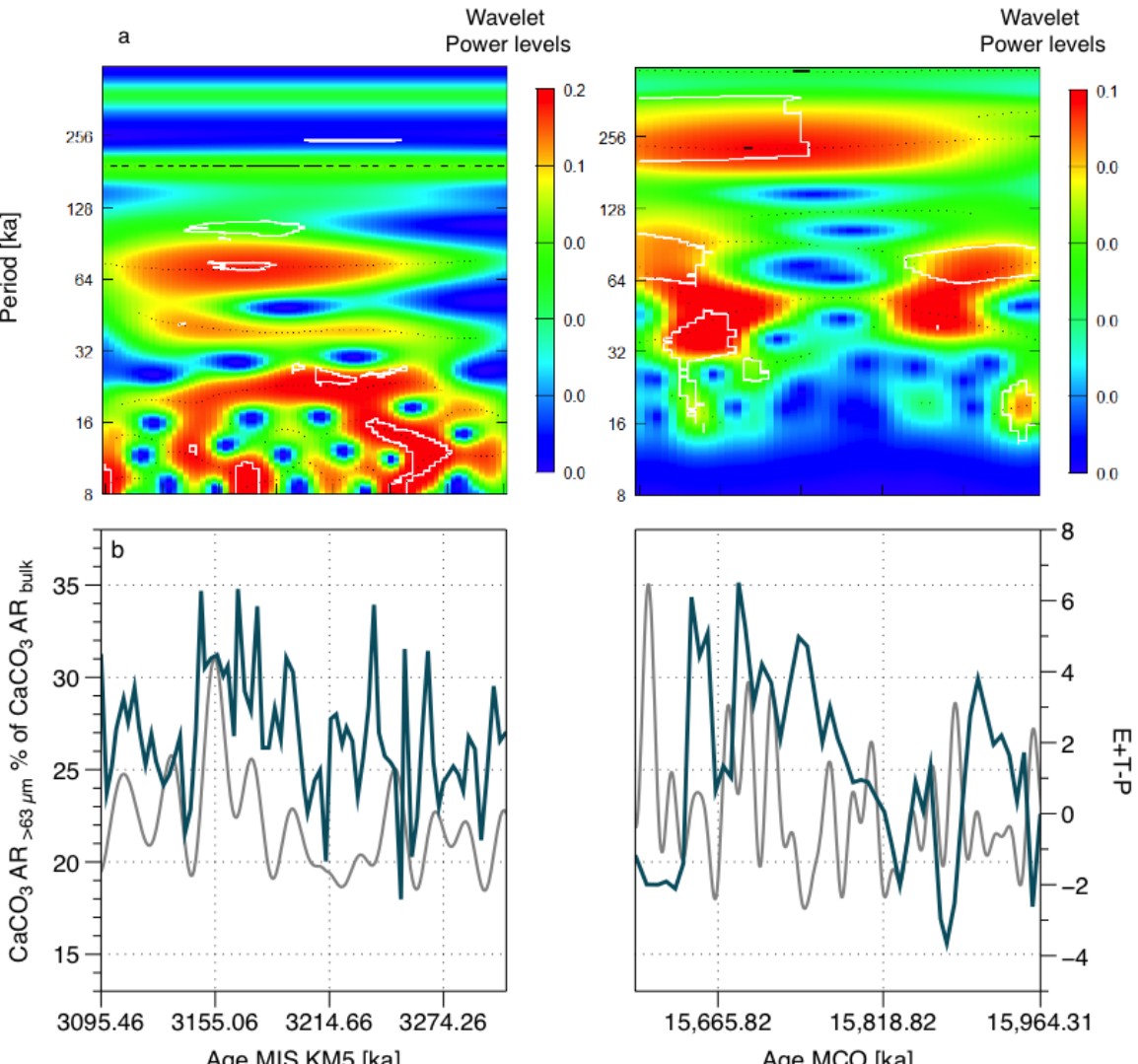

**Figure 3: a) Wavelet transform coarse fraction percentage of bulk CaCO₃ AR for both MIS KM5 and MCO,**
**significance value on the figure = 0.1, b) coarse fraction contribution to bulk CaCO₃ AR (solid blue line) and the actual**
**E+T-P (grey solid line) (Laskar et al., 2004).**

For the MIS KM5, we observe a clear and continuous precession imprint with a periodicity of approximately 20 ka between 3100 ka and 3270 ka, which appears to change to shorter cycles over time. In addition, we can observe an obliquity imprint (with a periodicity of approximately 41 ka), which plays a minor but not negligible role, and a clear eccentricity periodicity of 100 ka, which lies more or less between 3140 ka and 3230 ka. During the MCO, we see evidence of a 41 ka imprint (obliquity) from 15589 ka to 15900 ka. We cannot observe shorter periodicities because the resolution of the sampling does not allow this. The relative contribution of >63 μm carbonate to total $CaCO_3$ AR is in phase with E+T-P in the Pliocene and out of phase with E+T-P in the Miocene (as with total $CaCO_3$ AR, Cornuault et al., 2023) (Figure 3, 4 and supplementary figures S4 to S14).

We found 100 ka (eccentricity) periodicities in the >63 μm $CaCO_3$ AR and the <32 μm $CaCO_3$ AR (MIS KM5 and MCO), and the <32 μm $CaCO_3$ AR seems to be responding to 100 ka antiphased with a high AR when eccentricity is low and low AR when eccentricity is high (S1, S3). Additionally, we observe a 21 ka (precession) periodicity during the MIS KM5 and 41 ka (obliquity) periodicity during the MCO for the >63 μm $CaCO_3$ AR and the <63 μm $CaCO_3$ AR (S1, S2).

The >63 μm $CaCO_3$ AR and <32 μm $CaCO_3$ AR are in phase with E+T-P in the Pliocene, except during the peak of warmth of the PWP and antiphased during the MCO, suggesting that the coarse fraction (foraminifera) is responding in phase with E+T-P when the conditions are colder, and is in antiphase with E+T-P at a strong warm peak (PWP middle and all the Miocene interval chosen for this study). Even if the values are different, the variability is consistent between the <63 μm $CaCO_3$ AR (direct approach) and the <32 μm $CaCO_3$ AR (independent) SYRACO approach (Figure 2b).

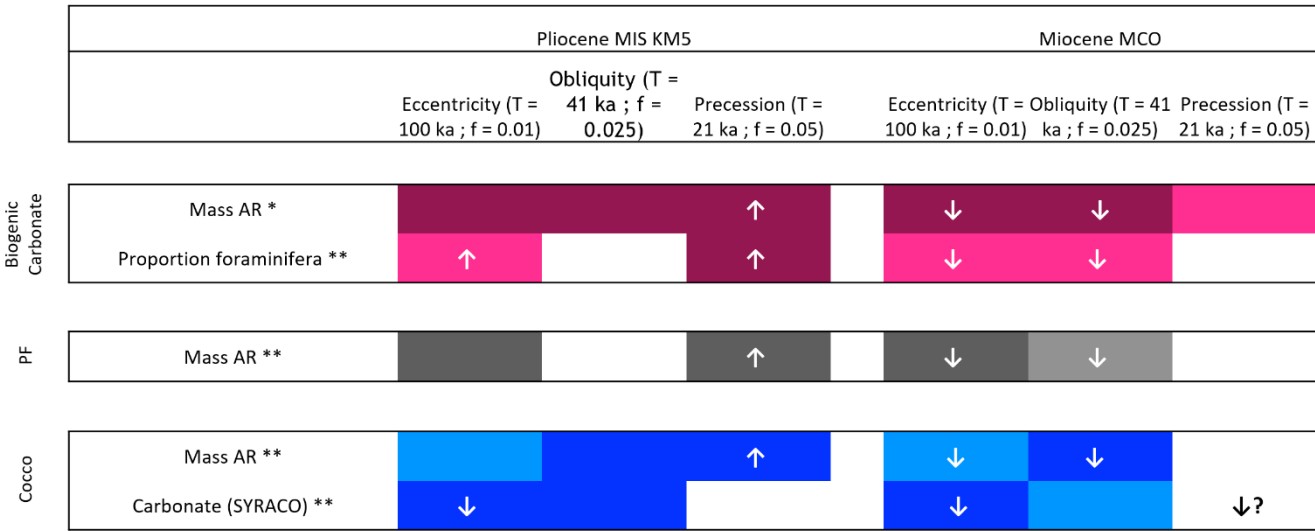

**Figure 4: Summary figure highlighting the key results from the spectral and wavelet analyses from Cornuault et al. (2023) and the present study regarding the total biogenic carbonate (pink), the planktonic foraminifera size fraction (grey) and the coccoliths size fraction (blue). The upward-pointing arrows are indicating that the tracked signal is in phase with the orbital signal, and the downward pointing arrows are indicating that the tracked signal is in antiphase with the orbital signal. \* Cornuault et al., 2023. \*\* This study.**

## 4 Discussion

There are possible factors influencing the different carbonate contents and carbonate accumulation of the various grain size fractions which could have potential confounding effects (e.g., selective dissolution, fragmentation, lateral transport) and therefore contradict the assumption that size-fraction partitioning reflects biological production differences between foraminifera and coccolithophores.

Because planktonic foraminifera and coccolithophores produce shells and plates respectively of very different sizes and shapes, they have a different sinking behaviour and susceptibility to dissolution. The coccoliths, up to two orders of magnitude smaller than foraminifera shells, sink quickly as they are often concentrated in aggregates and faecal pellets (Fischer and Karakaş, 2009; Richardson and Jackson, 2007; Ziveri et al., 2007). On the contrary, the foraminifera shells are composed of mesocrystalline material with a large surface-to-volume ratio, making their shells more susceptible to dissolution (Frenz et al., 2005; Honjo and Erez, 1978). This means that if we observe predominantly well-preserved foraminifera shells in our samples, the coccoliths should be mostly unaffected by dissolution as well. At this location, we know that the bottom waters became periodically affected by the corrosive Antarctic Bottom Waters only after the North Hemisphere Glaciations (Harris et al., 1997; Liebrand et al., 2016; Pälike et al., 2006), meaning that the samples from the Pliocene (MIS KM5) and the Miocene (MCO) should not be affected by dissolution. To support this statement, we generated in a previous study a fragmentation index of the planktonic foraminifera shells for these time intervals, a commonly accepted proxy to estimate the degree of carbonate dissolution of marine carbonate sediments (Berger et al., 1982; Preiss-Daimler et al., 2013), and found no significant effect of dissolution for the Pliocene and Miocene intervals (Cornuault et al., 2023). Comparing the fragmentation index with the carbonate record, we found no correlation, confirming the absence of dissolution (Cornuault et al., 2023). Furthermore, we also had a look to the samples below the binocular, and confirmed the absence of dissolution on our samples. The results of these analysis are published in Cornuault et al. (2023) (Fig S4 and S7). Therefore, we have various arguments at our disposal that allow us to assert the absence of a strong influence of dissolution during the Pliocene and Miocene periods. Previous studies and sediment cores observations of Site 927 show a very continuous record, mostly unaffected by slumps, turbidites and lateral transport (Cornuault et al., 2023; Curry et al., 1995), so we can consider that this source of disturbance can also be ruled out. Finally, there is a possibility that the total carbonate does not consist solely of coccoliths and (planktonic) foraminifera, but can also contain pteropods, and possibly benthic foraminifera and ostracods, etc., which can influence the total carbonate and coarse carbonate >63 µm. Since we do not observe much of them, we considered that the important variations in carbonate accumulation rates we do observe are not driven by changes of their carbonate productivity. Additionally, we are aware that coccoliths may not be the only formers of fine carbonate (<32 µm size fraction), fragments of foraminifera or even pteropods can also play a minor role here, as can perhaps also redeposited fine carbonate (from river or wind transport). In these conditions, the different class size carbonate accumulation rates should mainly reflect the biogenic production variability of the two main groups of pelagic carbonate producers.

There is an increase in the relative contribution of foraminifera (or decrease in the coccoliths contribution) from the Miocene to the Quaternary, which is consistent with the findings of Si and Rosenthal (2019). In addition, we observe different fluctuations in the relative contribution of foraminifera and coccoliths within the Pliocene and Miocene, indicating that the two most important carbonate producers react differently in these two periods. Furthermore, the phase relationship between the total CaCO$_3$ AR fraction and the contribution of foraminifera changed between the Pliocene and Miocene (Figure 1). For both time intervals, we do not observe any correlation between the contribution of the coarse fraction to the total pelagic carbonate production and the carbonate production itself (Figure 2), meaning that the changes in the relative contribution of the two main pelagic carbonate producers are not driving the changes in total CaCO$_3$ production. For the Pliocene and the Miocene, we observe large changes in the correlations between the different size fractions and the bulk CaCO$_3$ AR, and in addition a large amplitude variability of the coarse fraction contribution to the bulk CaCO$_3$ AR within these two time intervals, highlighting that the relative contribution of the two main pelagic carbonate calcifiers, while remaining similar on a geological time scale, has changed a lot on shorter orbital time scale.

For both the Pliocene and the Miocene, it is not the coarse fraction CaCO$_3$ production that is driving the CaCO$_3$ AR bulk changes, but the <63 µm CaCO$_3$ production. This conclusion was made by comparing our new records to those of Cornuault et al. (2023) on the basis of the variance showed by the reconstructed flux of the individual components and of the total carbonate. The total carbonate flux cannot be driven by variations in the coarse fraction flux, because this varies much less

than the total carbonate flux. This means that the two main pelagic carbonate producers responded differently, and that the coccolithophores relative contribution seems to be driving the overall changes. We found a coarse fraction contribution of about 20 to 30%, realistic and coherent with recent findings by Si and Rosenthal (2019) and Drury et al. (2021). The production within the two carbonate producers is changing synchronously and in the same direction, but with different amplitudes at both geological and orbital time scale, resulting in a change of their relative contribution to the $CaCO_3$ AR bulk.

According to the observations of the wavelet transform and $CaCO_3$ AR records vs time (Figures S1-S3), the foraminifera and the coccoliths are not responding to the same orbital parameter and not in the same way. The changes of $CaCO_3$ AR of the two groups do not seem to covary nor are they linked by any correlation, meaning that they are changing through the time without responding to the same forcing. This could also explain the different frequencies observed in the bulk $CaCO_3$ AR record (Cornuault et al., 2023): one frequency tracking coarse (foraminifera) and one frequency tracking small (coccolithophores) fraction. There is an equilibrium on long time scales between the two groups, so when the $CaCO_3$ AR bulk increases, both groups increase but not necessarily with the same amplitude (e.g. between the Miocene and the Pliocene). Indeed, it has been suggested that eccentricity changes in the surface ocean conditions drove coccolithophore evolution in the Pleistocene (Beaufort et al., 2022), whereas there is little evidence for evolutionary changes in planktonic foraminifera at such short time scales (Kucera and Schönfeld, 2007). An increase in eccentricity corresponds to an increase in the mean annual solar insolation at all latitudes (Goosse, 2015) and modulates the precession and seasonal insolation contrast, which is higher when the eccentricity is high. Nannoplankton evolution might be affected by orbital changes due to their dependence on light for photosynthesis and through the emergence of a greater diversity of ecological niches with higher seasonality (Longhurst, 2007), and so, the potential to carry a larger number of species, including larger or more calcified taxa (Beaufort et al., 2011; Henderiks and Bollmann, 2004).

Furthermore, some studies are highlighting the high sensitivity of the coccoliths production to eccentricity changes (e.g. Beaufort et al., 1997, 2022; Drury et al., 2021). The coccolithophores morphological diversity changes with the eccentricity modulation, as the result of coccolith evolution being forced by eccentricity, and so, affect the carbonate accumulation rate and drive carbon cycle changes (Beaufort et al., 2022).

We observe important differences of the orbitally forced variability within the different time intervals and different responses between the two groups. However, the carbonate production is one of the principal mechanisms for alkalinity removal from the surface ocean and the alkalinity has a strong influence on the ocean capacity to absorb $CO_2$ (Boudreau et al., 2018). As the two main pelagic calcifiers do not seem to be responding to the same forcing within each time interval, we can expect that if the conditions are changing, the ratio between the two groups will shift, affecting the ocean alkalinity and ocean capacity to absorb $CO_2$ (Boudreau et al., 2018). Our results are also in line with Holbourn et al. (2014), with a lower coarse fraction proportion centred in the MCO interval (corresponding to the lower $\delta^{18}O$ values). It is possible that above a certain temperature threshold (or $CO_2$ threshold) the response of the carbonate producers changes: as the contribution of the coarse fraction is in phase during the Pliocene and in antiphase during the Miocene, we suggest that above a certain temperature (or more generally, in some extreme environmental condition) the pelagic carbonate production (especially that of the planktonic foraminifera) drops. This could be explained if the ecological optima of the planktonic foraminifera and the coccolithophores were systematically different in some aspect (Beaufort et al., 2011; Schmidt et al., 2006), and the tropical species remained highly sensitive to this ecological aspect (Schmidt et al., 2006).

These processes are already being observed as shown by recent studies on modern coccolithophore and planktonic foraminifera. A decreasing calcium carbonate production of coccolithophores relative to growth with increasing pCO2 (and decreasing $CO_3^{2-}$ concentrations) has been found in most areas, with species-specific response (Gafar et al., 2018) and blooms seen to be limited by temperature increase (Oliver et al., 2024; Sauterey et al., 2023). For the future, coccolithophore dynamics are expected to be driven by ocean warming and stratification, as evidenced by their poleward shift in response to rising sea surface temperatures (D'Amario et al., 2020; Hutchins and Tagliabue, 2024; Meyer and Riebesell, 2015), and the

coccolithophore communities are described as varying with depth and latitude (Han et al., 2025). In recent years, high latitude regions are being potential refuges, with both coccolithophore and foraminifera expanding towards poles (Chaabane et al., 2024; Winter et al., 2014; Ying et al., 2024), but foraminifera acclimatization capacities are limited and insufficient to track warming rates and migration will not be enough to ensure survival (Chaabane et al., 2024). Models show species-specific responses of planktonic foraminifera shell weight changes with increasing $pCO_2$, associated with drivers including the carbonate system (Barrett et al., 2025). A weight reduction of some species has been observed between pre-industrial and post-industrial Holocene, evidence of a decrease in planktonic foraminifera calcification (Béjard et al., 2023). The extremely rapid responses of foraminifera suggest that climate change is already having a meaningful impact on coastal carbon cycling and the observed decrease in particulate inorganic carbon (PIC) flux relative to particulate organic carbon (POC) flux may facilitate increased oceanic uptake of atmospheric $CO_2$ (Havard et al., 2024). They are projected to reduce their global carbon biomass between 14% and 18% by 2100 as a response to ocean warming and associated changes in primary production and ecological dynamics. That decline is expected to slow down the ocean carbonate pump and create short-term positive feedback on rising atmospheric $pCO_2$ (Grigoratou et al., 2022; Ying et al., 2024). Foraminifera shell calcification has been shown to be responding to the anthropogenically driven reductions in ocean density, with potential consequences for the carbon cycle (Zarkogiannis et al., 2025). Certain species or strains may acclimate to future ocean pH and temperature (Rigual-Hernández et al., 2020; WestgÅrd et al., 2023), whereas others may perish or expand to new ecological niches where conditions are more suitable (Shemi et al., 2025), modifying the carbon export by the carbonate counter pump (Neukermans et al., 2023). Ziveri et al. (2023) observe a decoupling of $CaCO_3$ production and export, with pelagic $CaCO_3$ production being higher than the sinking flux of $CaCO_3$ at 150 and 200 m, implying the remineralisation of large portion of pelagic calcium carbonate within the photic zone. The relative abundance of foraminifera to coccolithophores has been highlighted as an important factor that could lead to large changes in the amount of $CaCO_3$ exported from the surface ocean and thus the cycle of alkalinity (Ziveri et al., 2023). A decrease in the role of the coccoliths in the surface ocean anthropogenic carbon absorption in the future with warmer climates would reduce the Ocean's carbon sink capacity (Ziveri et al., 2023).

## 5 Conclusion

Our results indicate that in the tropical Atlantic, since the Miocene, the contribution of coccoliths and planktonic foraminifera as the two important pelagic carbonate producers has varied on orbital time scales, but remained similar on average on longer time scales. We observe that the changes in the contribution of the coarse fraction to the total pelagic carbonate production followed mainly precession in the Pliocene, and obliquity and eccentricity in the Miocene and that the phase relationship to orbital forcing appears to have shifted between the two intervals. Individually, we show that the coarse fraction $CaCO_3$ AR was responding to precession and obliquity and the small fraction $CaCO_3$ AR was responding to eccentricity, indicating that the two pelagic carbonate calcifiers were responding to different orbital forcing, and following different phase relationships. It remains unknown whether the observed changes in their contribution to total carbonate reflect differences in population growth or changes in the size or calcification intensity of their skeletal elements. Either way, it appears that eccentricity not only modulates coccolithophore community assembly and evolution (Beaufort et al., 2022), but also their carbonate production, affecting the composition of pelagic carbonate deposited on the seafloor. In this way, the hitherto overlooked variability in coccolithophore productivity and foraminifera productivity may play a role in the global carbon cycle. This is because the calcite skeletons of the two groups are associated with different amounts of organic carbon, they contain different amounts of trace elements and they sink through the water column following different physics resulting in different ballasting effects on the biological carbon pump. Because coccolithophore and foraminifera calcite have different susceptibility to dissolution, a large change in the carbonate production by either of these two calcifiers could have an impact on the ocean capacity to regulate atmospheric $CO_2$ (Boudreau et al., 2018). Indeed, if their carbonate production changes, the quantity of

carbon exported to the deep ocean also changes. As pelagic calcifiers, they are important actors in the ions removal from the waters of the surface ocean and so, alkalinity removal. This alkalinity is strongly regulating the capacity of the ocean to absorb atmospheric $CO_2$ via gas exchanges at the surface of the ocean (Boudreau et al., 2018).

## 6 Data availability

The datatable corresponding to this manuscript are available on PANGAEA (https://pangaea.de):

Cornuault, Pauline; Beaufort, Luc; Pälike, Heiko; Bickert, Torsten; Baumann, Karl-Heinz; Kucera, Michal (2025): Coarse and small fraction carbonate content and accumulation rate from Miocene to Quaternary, at ODP Site 154-927, Ceara Rise, tropical Atlantic [dataset]. PANGAEA, https://doi.org/10.1594/PANGAEA.974795.

This datatable contains the samples list, their ages and the mass of <32 µm sediment on the lamellas for coccoliths analyses, the mass of carbonate on these slides (measured with SYRACO automated device), the dry weight of the bulk sediment, the weight of the >63 µm size fraction, the accumulation rate of the >63 µm size fraction, and the accumulation rate of the <32 µm size fraction.

## 7 Author contribution

PC, the first author, is the main contributor to the manuscript. She prepared the research project, ran most of the experiments in MARUM (Bremen, Germany), interpreted the data, prepared all the figures, interpreted the data and wrote this manuscript. MK is the principal investigator, and contributed a lot in the writing of this manuscript. HP, TB and KHB helped with the data interpretation and the writing of this manuscript.

LB ran the analysis of the small fraction samples in CEREGE (Aix-en-Provence, France) and helped with the data analysis and the writing of the present manuscript.

## 8 Competing interests

All the authors declare no conflict of interest for this work.

## 9 Acknowledgments

This research used samples and data provided by the Ocean Drilling Program (ODP), sponsored by the US National Science Foundation (NSF) and participating countries. This research was supported by the Deutsche Forschungsgemeinschaft (DFG) through the Cluster of Excellence "The Ocean Floor - Earth's Uncharted Interface" (EXC-2077, Project 390741603). A special thanks to the staff of the Beaufort lab who helped with the coccolith analysis at CEREGE using SYRACO, at a time when it was impossible to travel due to the COVID-19 pandemic.

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
