# Peer review of "Orbital-scale variability in the contribution of foraminifera and coccolithophores to pelagic carbonate production"

_EGUsphere, 2025_

## Referee Comment (RC1)

**Cornault et al. 2025 – Orbital-scale variability in the contribution of foraminifera and coccolithophores to pelagic carbonate production**

*Summary*

This manuscript explores orbital-scale variation in the relative contribution of foraminifera and coccolithophores to the pelagic carbonate burial flux at a single site in the tropical Atlantic during the Miocene, Pliocene and Quaternary. The authors use two independent methods to identify the proportion of total carbonate composed of foraminifera tests vs coccoliths, weighing the biases of each approach. They apply spectral analysis to look for orbital patterns and longer term trends, and they check for covariation between carbonate accumulation rate and the relative contribution of foraminifera vs coccolithophores. Finally, they consider their results in terms of temperature thresholds, light requirements, evolution, and ecological optima.

The data presented here is new and the manuscript is likely to be interesting to a relatively broad audience, as it represents progress in understanding what can be gleaned from the relative contributions of forams vs coccoliths as well as understanding the response of carbonate production and burial to drivers on both million-year and orbital timescales. Further, as the data presented here coincides with previously collected data from the same site and time intervals, it also allows for better interpretation of what was happening at that particular site (and in the tropical Atlantic generally) during the last 15 My. The manuscript is clear in its focus and doesn't make excessive or overgeneralized claims, and the use of two different methods with very different biases alleviates challenges to interpretation and creates a more holistic picture (although these could be easily bolstered with a simple description of the actual sediment during measurement, rather than simply assuming its status – see details in the line-by-line).

As far as the data collection and overall presentation goes, I would recommend publication with minor edits. However, the discussion and interpretation of the data falls far short of its potential, and leaves much to be desired. In part this is due to confusingly worded sentences and poorly organized paragraphs in the discussion, which make it very challenging to follow. Additionally, pieces of the interpretation are left too vague for the reader to be able to properly consider the viability of the arguments made. It isn't clear, for instance, why a larger number of species should make nannoplankton more susceptible to eccentricity, or how more rapid coccolithophore evolution would 'drive carbon cycle changes'. There is oblique reference to how variation in the relative abundance of forams or coccolithophores would affect the ocean's carbon sink capacity, but these are never explained in any detail. I would also love to see more done (or done more clearly) with the overall results of Figure 2, concerning the relationship between relative calcifier abundances and total carbonate accumulation rate, as this seems to be ideal data for understanding how orbitally induced environmental changes affect carbonate burial. This is addressed in the discussion, but in such a confusing way that I wasn't able to get much out of it.

One other persistent issue is the dismissal of some pieces of the data to 'the susceptibility of foraminifera and coccolithophores to carbonate dissolution', without any explanation of why

this is believed to be the case, let alone the carefully constructed and well-evidenced argument that such an interpretation deserves (I have identified each instance of this problem in the line-by-line below). More generally, dissolution would be worth addressing with some thoroughness, as the *relationship* between carbonate production and burial is susceptible to the same environmental factors that govern carbonate production.

Overall, this paper has high potential and presents valuable data, and everything up to the discussion section looks more or less great, but its discussion needs rewriting. I recommend publication contingent on its discussion being overhauled and the above issues being addressed. Given the vast experience of the senior authors on this manuscript in precisely this field, I am honestly somewhat surprised that the manuscript was submitted in this state. The above issues are ones that should have been identified and attended to by the senior authors (who are more than capable of addressing them) prior to submission, rather than work done by reviewers. I hope to see this manuscript revised with more attention from the senior authors, who are in a better position to help develop the lead author's writing and will benefit more from authorship than any reviewer will from putting in this work.

**Line-by-line**
*The below section identifies a few additional instances of confusion on the scientific points made as well as minor issues of grammar or suggestions for better readability. The English (outside of the discussion section) is superb, but a final read-through of the subsequent draft by someone even more versed in academic English could lend it extra polish.*

L18...: neithers and nors should be eithers and ors; 'orbital parameter modulations'; last bit of abstract should be 'deposition on either geological or orbital time scales'.

L37 and throughout: 'calcareous nannoplankton' (nannoplankton pluralized without the s)

L48: what about their likelihood of being in faecal pellets, or the amount of cell material covering them? Seems like there could be more than three factors, and there's no citation to indicate why these three are even thought to be the most critical factors.

L53: 'long geological time scale' could be clarified here with a timeframe – thousands? Millions of years?

L53: 'less' should be 'fewer'

L54: 'explain it to be due to' might be better worded as 'attribute it to'

L52-54: This whole sentence is quite confusing, actually, and I'm not sure what it's saying. Is the decrease in weathering alkalinity? What direction is the CO2 modulation?

L57: main (no s); estimated from **the** carbonate accumulation

L61: development of **a** tuned age model and quantification of **the** flux; also this sentence could do with being broken up into two sentences

L71: mbsl (for consistency with its use later)

L72: episodes **have** resulted

L76-77: I'm confused about 'show no relationship between carbonate content and carbonate flux'... wouldn't there be other reasons besides shoaling to have a relationship between % carbonate and the total carbonate mass accumulation? Changes in silicifier productivity, for instance?

L79: 'at different time scales' could be clarified

L80: Given its importance to the findings, it might be helpful to the reader to briefly summarize here by what method we know the carbonate content (so they don't have to track it down in Cornault et al 2023).

L88: 'In the first approach...' might be a less ambiguous way to word it than 'First' (which sounds like the first part of an approach)

L89: What else is there? What proportions? And would it make sense to have this sentence in the first section of the methods, with the rest of the site overview information?

L92-93: It might be helpful to your reader to elaborate on this, given how relevant it is to the method.

L94: 'in tap water in 15 ml centrifuge tubes in a rotating carousel'

L95: 'dry bulk sediment (DBS) weight'

L95: washed... and dried, presumably? How?

L90-103: Did you check (visually) to see if the DBS was carbonate only (ie not much contribution from silicifiers, clay mostly removed), and that the <63 fraction was almost entirely foram? You say you assumed it, but it seems like an easy thing to check, at least roughly, and it would support your use of the method.

L110-111: 'centrifuge tube'

L~120: It might be useful to provide a rough estimate of coccolith and foram size ranges during this time/at this site, for your readers who are less familiar with the subject.

L126: 'these changes' haven't been introduced yet – need to be introduced before they can be referred to like this.

L128:…'not for the Quaternary, as these two time intervals are…' – maybe it's just me but I found this confusing.

L129: 'the change' should be 'changes'

L131: E+T-P should be defined for your readers who are less familiar with the subject.

L135: black **solid** line

L135-140: I'm confused… the vertical blue lines don't seem to correspond to the highest solid line values in the third and fourth plots (MIS KM5 and MCO)?

L143-144: 'which is related to the susceptibility of foraminifera and coccolithophores to carbonate dissolution' – this is interpretation! Doesn't seem like it belongs in the Results section, and also there's no evidence given to back the claim. Why do you think that's the case?

L146: Is there a reason to keep Fig S3 in the Supplement? It seems like it would be helpful in the Main Text. It would also be good to state somewhere in the Main Text how many samples were analyzed.

L147: peak values… are these the average? I was confused by the way this was worded

L149: 'consistent with the effect of dissolution'… based on what?? This keeps getting stated without any citation or explanation. The wording here is also a bit odd – maybe these could be broken into separate observations? Or I'm confused about what it's trying to say.

L155: 'is showing' should be 'shows'

L156-… 'During the Quaternary…' – this is interpretation!! It should be moved to the Discussion section.

L159: 'taking in account maxima values average' should be (if I'm understanding correctly) 'using the averages of the maximum values'

Fig. 2b: I'm curious if the slopes vary by site. Also, maybe instead of the three modeled regression lines, you could provide just the slope and y-intercept of the actual line? Just a thought, not necessary.

L170: Seems like this should come before L150?

L171-172: Again with the unsupported assertion about carbonate dissolution!! This needs to be solidly explained the first time with evidence, citations, and spelled-out logic if it's going to be regularly referenced and relied upon for later argument.

L176: using **the** SYRACO device

L178: expect **them to be correlated with** the SYRACO **values systematically** lower **and no** differences…

L177-179: This isn't highlighted by the figure, but it would be good if it were.

L190: 'Compared to…' – I found this sentence confusing, and I'm not sure how best to reword it to make it less ambiguous.

L193: would **not** allow; contribution of the >63 µm **fraction** to the bulk

L204: Fig. 2b doesn't show this super well… maybe it would be helpful to have a figure more akin to Figure 1 (or a 2$^{nd}$ panel) showing it?

L206: increase **in** the contribution of foraminifera **(or decrease in the coccolithophore contribution) from the Miocene to the Quaternary** coherent with

L207: say what Si and Rosenthal found, so your reader doesn't have to track down the paper

L 207-210: The sentence beginning 'Furthermore' is very confusing. Different between the Pliocene and the Miocene? Or different within the Quaternary? Also the 'so' would perhaps be better written as 'indicating that'

L210: it **changes** between; For **these two** time intervals we do not; -- this is actually confusing because then you're talking about Quaternary cold events… so are we still talking about the Pliocene and Miocene? I'm lost.

L214: '…proof of the preferential dissolution…' you're arguing it constitutes 'proof' based on not a single scrap of evidence or explanation! The rest of the manuscript seems carefully constructed to build arguments based on observation, so I'm not sure what the blind spot is about this one thing. But it needs fixing.

L217: carbonate calcifiers, **while remaining** similar on **a** geologic time scale

L218: 'it isn't' should be **it is not** (don't usually use contractions in formal writing)

L219: CaCO3 production **that** is driving the

L218-219: How does this work with the absence of a correlation?

L221: remove the comma after (2019)

L223-224: time scale, **producing changes in their** relative contribution to the CaCO3 AR bulk**.**

L227: covary **nor are they** linked

L227-228: Point 1 seems a bit repetitive? They're not linked, meaning they're not linked…

L231: AR bulk **increases, both groups increase but not necessarily to the same degree** (e.g.

L233: increase **in** eccentricity corresponds to an increase **in** the mean

L235: due to **their** dependence

L236: Why would a larger number of species cause them to be more eccentricity affected? This could do with elaboration, as could the evolutionary forcing of coccolithophores by eccentricity (L239) and how this affects the carbonate accumulation rate and drives carbon cycle changes. This is the part of the meat of the discussion, and these are not self-evident claims that the broad readership of this journal will necessarily be familiar with. In addition to citing the relevant papers it would greatly help the discussion if these interpretations were explained or pieced together in the main text. These could have their own paragraphs even; there's plenty of space.

L239: as the **result** of; **coccolithophore** evolution (or **coccolith**, but not coccoliths)

L240: I would find it easier to read this section if there was a paragraph break after (Beaufort et al., 2022).

L241: As the two **main** pelagic

L244: biomineralization process... This could be more specific, and elaborated on.

L246: so if **both foraminifera and coccolith** productivity (or **coccolithophore** might be more accurate?)

L245-246: Are we talking about the PWP? Otherwise I'm lost again, since there's purportedly no correlation...

L248-250: Way more is needed here to flesh this out (and it could be interesting to explore!) and the English needs some work as well, and the organization (why does the last sentence happen there?).

L253-260: Same with this paragraph.

L275: trace elements**,** and **they sink** through the water column

L277: a large change **in** the

L277: 'by these two calcifiers'... by one rather than the other, you mean, presumably?

L282: of **>32 μm** sediment

L289-293: Surely you're not thinking of redacting it already? ☺

---

## Author Comment (AC1)

**Response to reviewer 1**

We thank the reviewers for their thoughtful and constructive comments, which will help us improve the quality of our manuscript. Below we respond to each point raised, and indicate when changes will be made in the manuscript. Reviewer's comments are in Italic font and our responses are plain text.

*It isn't clear, for instance, why a larger number of species should make nannoplankton more susceptible to eccentricity, or how more rapid coccolithophore evolution would 'drive carbon cycle changes'. There is oblique reference to how variation in the relative abundance of forams or coccolithophores would affect the ocean's carbon sink capacity, but these are never explained in any detail. I would also love to see more done (or done more clearly) with the overall results of Figure 2, concerning the relationship between relative calcifier abundances and total carbonate accumulation rate, as this seems to be ideal data for understanding how orbitally induced environmental changes affect carbonate burial. This is addressed in the discussion, but in such a confusing way that I wasn't able to get much out of it.*

*More generally, dissolution would be worth addressing with some thoroughness, as the relationship between carbonate production and burial is susceptible to the same environmental factors that govern carbonate production.*

We thank the referee for the suggestion to extend the discussion on: a) the link between orbital variability and coccolithophore evolution as a potential mechanism driving the changes in pelagic carbonate composition, b) the link between pelagic carbonate composition and ocean carbon cycle and b) the reasons why we believe that dissolution is not driving the observed cycles.

This will be taken in account by adding, among other things, a paragraph in the discussion including recent references of modern observations regarding both coccolithophores and foraminifera, their ecology and their current carbonate production during the ongoing climate warming.

**Line-by-line**

*L18…: neithers and nors should be eithers and ors; 'orbital parameter modulations'; last bit of abstract should be 'deposition on either geological or orbital time scales'.*

This will be changed in the revised version of the manuscript.

*L37 and throughout: 'calcareous nannoplankton' (nannoplankton pluralized without the s)*

This will be changed in the revised version of the manuscript.

*L48: what about their likelihood of being in faecal pellets, or the amount of cell material covering them? Seems like there could be more than three factors, and there's no citation to indicate why these three are even thought to be the most critical factors.*

There is indeed a subtle mistake in this sentence, which the referee correctly spotted. The three factors that we mention are responsible for the amount of pelagic carbonate that is leaving the productive zone, not "arriving on the seafloor". We will change the sentence accordingly.

Additionally, the third point in the text will be explained a bit more extensively, and 'their investment in biomineralisation relative to cell size' will be detailed as being the thickness and density of the shell regarding the size of the organism. However, we would like to point out that faecal pellets are

considered one of the main transport mechanisms for the sinking of coccoliths and therefore certainly play a role in transport to the sea floor. However, these seem to disintegrate completely in the water column or, at the latest, in the sediment, as they are not or very seldomly observed in the sediment or even in sediment traps. Furthermore, zooplankton faecal pellets have long been thought to be a dominant component of the sedimentary flux in marine ecosystems, but many studies using sediment traps have shown that they often constitute only a minor or variable proportion of the sedimentary flux (e.g., Turner, 2002). It is not entirely clear to us what is meant by 'the amount of cell material covering them'. Our database is based on precise counts that are not influenced by any overlaps or materials on top of the carbonate plankters.

Turner, J.: Zooplankton fecal pellets, marine snow and sinking phytoplankton blooms, Aquat. Microb. Ecol., 27, 57–102, https://doi.org/10.3354/ame027057, 2002.

*L53: 'long geological time scale' could be clarified here with a timeframe – thousands? Millions of years?*

Thank you for pointing it out, this will be modified in the revised version of the manuscript as follow: "[…] long geological time scale (Millions of years) […]"

*L53: 'less' should be 'fewer'*

This will be modified in the final version of the manuscript.

*L54: 'explain it to be due to' might be better worded as 'attribute it to'*

This will be reworded in the revised version as "[…] and attribute it to long-term […]".

*L52-54: This whole sentence is quite confusing, actually, and I'm not sure what it's saying. Is the decrease in weathering alkalinity? What direction is the CO2 modulation?*

Thank you for pointing this out. We plan to update this sentence to make it clearer, and plan to reword it as follow: "Si and Rosenthal (2019) show a long geological time scale shift towards more foraminifera and proportionally less coccoliths towards the Quaternary. They attribute it to a long-term decrease in ocean alkalinity supplied by continental weathering, due to decreasing pCO2 throughout this time interval."

*L57: main (no s); estimated from **the** carbonate accumulation*

This will be modified in the revised version of the manuscript.

*L61: development of **a** tuned age model and quantification of **the** flux; also this sentence could do with being broken up into two sentences*

Thank you for these suggestions, the revised manuscript will be as follow: "To understand to what degree the composition of pelagic carbonate flux varied in the past and to qualify and quantify the changes in the differential contribution of the two mains pelagic calcifiers to the total pelagic carbonate production estimated from carbonate accumulation rate (CaCO3 AR) (Brummer and van Eijden, 1992; Liebrand et al., 2016), we generated new data for Leg 154 ODP Site 927, Ceará Rise. This Site was chosen for its good carbonate preservation during Quaternary interglacials and throughout the Neogene (Curry et al., 1995) and because the site remained in a tropical setting throughout, while

at the same time, well-recorded orbital cyclicity of sediment properties allow the development of a tuned age model and quantification of the flux at orbital resolution."

*L71: mbsl (for consistency with its use later)*

This will be modified in the final version of the manuscript

*L72: episodes **have** resulted*

We will change it for the final version of the manuscript

*L76-77: I'm confused about 'show no relationship between carbonate content and carbonate flux'… wouldn't there be other reasons besides shoaling to have a relationship between % carbonate and the total carbonate mass accumulation? Changes in silicifier productivity, for instance?*

Yes, this is correct, which is why we list three factors in the same sentence, which we collectively interpret as indicative for little role of carbonate dissolution in shaping the composition of the sediment. To make this clear, we will expand this sentence, explaining how each of the three factors supports the collective claim.

The argument here should prove that the sediments in the intervals studied are not influenced by carbonate dissolution. Therefore, other causes must of course be responsible for the mismatch between carbonate content and carbonate flux (= carbonate accumulation). We will emphasise this more clearly in the revision. The distribution of diatoms (as the main silicifier) in sediment deposits, and the factors that control their distributions, are often discussed by researchers when reconstructing both contemporary environments and paleoenvironments. However, they are more important in higher productive upwelling areas or at higher latitudes and the fossil record of diatoms is also strongly influenced by changes in preservation potential so that does not necessarily reflect a primary ecological signal (e.g., Westacott et al., 2021). Furthermore, it is difficult to 'convert' the silicate content into masses, and absolute silicate measurements are rare. We have therefore neglected them in our reconstructions carried out here.

Westacott, S., Planavsky, N. J., Zhao, M.-Y., and Hull, P. M.: Revisiting the sedimentary record of the rise of diatoms, Proc. Natl. Acad. Sci. U.S.A., 118, e2103517118, https://doi.org/10.1073/pnas.2103517118, 2021.

*L79: 'at different time scales' could be clarified*

This will be modified as follow: […] at time scales from thousand years to million years."

*L80: Given its importance to the findings, it might be helpful to the reader to briefly summarize here by what method we know the carbonate content (so they don't have to track it down in Cornuault et al 2023).*

We thank the reviewer for this suggestion, we will add a sentence to address it as follow: "The carbonate content analyses on the bulk sediment were done using a LECO CS744 elemental analyser at Bremen University."

*L88: 'In the first approach…' might be a less ambiguous way to word it than 'First' (which sounds like the first part of an approach)*

Sure, it will be changed in the revised version of the manuscript.

*L89: What else is there? What proportions? And would it make sense to have this sentence in the first section of the methods, with the rest of the site overview information?*

At the studied location, a very low portion of the carbonate content is composed of shell fragments as mentioned by Curry et al. (1995). We think that this sentence belongs to this section describing the carbonate content, as it is an important sentence for the methods we describe right after.

*L92-93: It might be helpful to your reader to elaborate on this, given how relevant it is to the method.*

Yes, we will add a sentence after as follow: "The remaining small portion of the foraminifera shell are from juveniles".

*L94: 'in tap water in 15 ml centrifuge tubes in a rotating carousel'*

This will be modified in the final version of the manuscript.

*L95: 'dry bulk sediment (DBS) weight'*

We will change it for the final version of the manuscript.

*L95: washed… and dried, presumably? How?*

We plan to add a sentence right after as follow: "The coarse fraction was then dried at 50°C overnight."

*L90-103: Did you check (visually) to see if the DBS was carbonate only (ie not much contribution from silicifiers, clay mostly removed), and that the <63 fraction was almost entirely foram? You say you assumed it, but it seems like an easy thing to check, at least roughly, and it would support your use of the method.*

Of course, the samples were visually checked before stating that it was carbonate only and that the coarse fraction was entirely foram.

*L110-111: 'centrifuge tube'*

*This will be modified in the final version of the manuscript.*

*L~120: It might be useful to provide a rough estimate of coccolith and foram size ranges during this time/at this site, for your readers who are less familiar with the subject.*

It will be added in the revised version of the manuscript.

*L126: 'these changes' haven't been introduced yet – need to be introduced before they can be referred to like this.*

We thank the reviewer for noticing this, we will modify the sentence as follow for the revised manuscript: "To see whether the CaCO3 AR changes of each size fraction are orbitally driven or not, […]"

*L128:…'not for the Quaternary, as these two time intervals are…' – maybe it's just me but I found this confusing.*

This will be modified as "[…] (except for the Quaternary, […]"

*L129: 'the change' should be 'changes'*

This will be modified in the revised version of the manuscript.

*L131: E+T-P should be defined for your readers who are less familiar with the subject.*

We thank the reviewer for this helpful comment, it will be added in the sentence as follow: "[…] our results to an E+T-P (eccentricity plus tilt minus precession) record (Laskar et al., 2004) […]".

*L135: black **solid** line*

It will be modified for the revised version of the manuscript.

*L135-140: I'm confused… the vertical blue lines don't seem to correspond to the highest solid line values in the third and fourth plots (MIS KM5 and MCO)?*

No, it's normal, it is because we took the highest values of the >63 μm carbonate contribution to the total CaCO3 (background fill). We will update the text to make it clearer L137 to 138, stating "Representative values (solid black lines with numbers) of the contribution of the coarse fraction to the bulk carbonate accumulation rate were calculated for each of the four time intervals as means of the two highest coarse contribution values (highlighted by vertical blue lines) in each interval."

*L143-144: 'which is related to the susceptibility of foraminifera and coccolithophores to carbonate dissolution' – this is interpretation! Doesn't seem like it belongs in the Results section, and also there's no evidence given to back the claim. Why do you think that's the case?*

We thank the reviewer for this feedback and plan to move the sentence to the discussion section. As to the basis of the claims made in this sentence, there is substantial body of literature supporting our statement. We will elaborate and cite it in the discussion. For example, it has been shown that foraminifera are more sensitive to dissolution than coccoliths (Honjo and Erez, 1978; Frenz et al., 2005). and studies found that the depth of the CCD has moves over the Quaternary glacials and that a shallower CCD is at the origin of carbonate dissolution Curry et al., 1995; Gröger et al., 2003b; Frenz et al., 2006). At this location, accumulation rate mainly reflects changes in pelagic carbonate production (Brummer and van Eijden, 1992; Bassinot et al., 1997).

*L146: Is there a reason to keep Fig S3 in the Supplement? It seems like it would be helpful in the Main Text. It would also be good to state somewhere in the Main Text how many samples were analyzed.*

We do think that keeping S3 in the supplements keeps the text light. Sure, we will add in the methods a sentence stating how many samples (261) were analysed.

*L147: peak values… are these the average? I was confused by the way this was worded*

No, the peak values are referring to the peaks (highest values shown by vertical bars in Figure 1) we observe in the record of foraminifera size fraction contribution to the total CaCO3 AR. We will reword that sentence to make it less confusing: "In contrast to the strong and consistent variability on orbital

time scales, the highest values of foraminifera fraction contribution for the four intervals studied were remarkably similar, ranging between 31.3% for MIS 9 and 39.8% for MIS 5 (Figure 1)."

*L149: 'consistent with the effect of dissolution'… based on what?? This keeps getting stated without any citation or explanation. The wording here is also a bit odd – maybe these could be broken into separate observations? Or I'm confused about what it's trying to say.*

We mentioned at the beginning of the Material and methods, L72 to L76 "However, the depth of the lysocline has varied in the past, and some of the shoaling episodes has resulted in the site being affected by carbonate dissolution. Such episodes are known from the glacial periods of the late Quaternary and are related to the restructuring of the Atlantic Meridional Overturning Circulation (AMOC) and an increased influence of the more corrosive Antarctic bottom waters (Gröger et al., 2003b). During the studied intervals of the Pliocene and Miocene, no such events occurred at this site."

We realise that this statement is unclear. What we mean is that the fact that the peak values during the Quaternary show carbonate accumulation and composition values that are comparable to those of the older intervals is compatible with previous inference that these intervals of the Quaternary represent times when the sediment was not affected by dissolution. We agree that this requires explanation and we will therefore move this section to the discussion and elaborate, as suggested by the referee below.

The wording is certainly not very successful and will be changed as follows: "The maximum values of the carbonate fraction of foraminifera for the four intervals studied are quite similar, ranging from 31.3% for MIS 9 to 39.8% for MIS 5 (Figure 1), and fit very well with the strong positive correlation between foraminifera fraction contribution to the total CaCO3 AR and bulk carbonate flux in the Quaternary intervals (Figure 2). This consistency contrasts with the strong and consistent variability on orbital time scales."

*L155: 'is showing' should be 'shows'*

We will modify it for the revised version of the manuscript.

*L156-… 'During the Quaternary…' – this is interpretation!! It should be moved to the Discussion section.*

This sentence will be moved to the Discussion section in the final version of the manuscript.

*L159: 'taking in account maxima values average' should be (if I'm understanding correctly) 'using the averages of the maximum values'*

It is indeed what we wanted to say, we will modify the sentence in the final version of the manuscript.

*Fig. 2b: I'm curious if the slopes vary by site. Also, maybe instead of the three modeled regression lines, you could provide just the slope and y-intercept of the actual line? Just a thought, not necessary.*

All the samples of this study are from the same site. This only makes sense to do it for the Quaternary and the Miocene+Pliocene, but not for all together. We either add the overall regression or state what it shows in the text (maybe if we add it, it makes the figure hard to read, for this, we need to see the figure with a common regression line). We will consider adding the equation of the actual line on the figure or legend.

*L170: Seems like this should come before L150?*

This will be moved for the revised version of the manuscript. The order of the arguments here will change, because we will move the section around Line 150 to the discussion.

*L171-172: Again with the unsupported assertion about carbonate dissolution!! This needs to be solidly explained the first time with evidence, citations, and spelled-out logic if it's going to be regularly referenced and relied upon for later argument.*

The assertion about the absence of dissolution affecting these intervals is supported, but we understand that the referee wishes this to be stated clearly and therefore, as also explained above we will dedicate a longer section to this argument, with references and a correspondingly clear and well-founded explanation, as requested.

*L176: using **the** SYRACO device*

This will be taken in account for the final version of the manuscript.

*L178: expect **them to be correlated with** the SYRACO **values systematically** lower **and no** differences…*

This will be modified in the final version of the manuscript.

*L177-179: This isn't highlighted by the figure, but it would be good if it were.*

We will consider to what extent we can highlight in the figure what has been said.

We thank the referee for spotting that in our listing we forgot to provide an information on the third argument: "that there are no differences in the shape of the relationship among the four studied intervals". We have evaluated this by calculating regressions separately for each of the four intervals, showing that they are similar (but not identical) and we will provide and discuss the results in the revised version.

*L190: 'Compared to…' – I found this sentence confusing, and I'm not sure how best to reword it to make it less ambiguous.*

We either reword it as follow: "For the MIS KM5, we observe a clear and continuous precession imprint with a periodicity of approximately 20 ka between 3100 ka and 3270 ka, which appears to change to shorter cycles over time. In addition, we can observe an obliquity imprint (with a periodicity of approximately 41 ka), which plays a minor but not negligible role, and a clear eccentricity periodicity of 100 ka, which lies more or less between 3140 ka and 3230 ka. During the MCO, we see evidence of a 41 ka imprint (obliquity) from 15589 ka to 15900 ka. We cannot observe shorter periodicities because the resolution of the sampling does not allow this. The relative contribution of >63 µm carbonate to total $CaCO_3$ AR is in phase with E+T-P in the Pliocene and out of phase with E+T-P in the Miocene (as with total $CaCO_3$ AR, Cornuault et al., 2023) (Figure 3 and supplementary figures S4 to S14)." or delete it.

*L193: would **not** allow; contribution of the >63 µm **fraction** to the bulk*

This will be modified in the revised version of the manuscript.

*L204: Fig. 2b doesn't show this super well… maybe it would be helpful to have a figure more akin to Figure 1 (or a 2$^{nd}$ panel) showing it?*

It is indeed fair to request a figure showing the temporal variation of the two proxies directly. Adding this to Figure 2 would make the plot overloaded, so we propose to provide this plot in the supplement.

*L206: increase **in** the contribution of foraminifera **(or decrease in the coccolithophore contribution) from the Miocene to the Quaternary** coherent with*

This sentence will be modified taking in account the suggestions of the reviewer.

*L207: say what Si and Rosenthal found, so your reader doesn't have to track down the paper*

We understand the reviewer concern, but since it was already written in the introduction L33,L34 and L52 to 54, we made the choice of not repeating it here.

*L 207-210: The sentence beginning 'Furthermore' is very confusing. Different between the Pliocene and the Miocene? Or different within the Quaternary? Also the 'so' would perhaps be better written as 'indicating that'*

We agree that this sentence is hard to understand. This is likely because it is too long and the individual statements lack explanation. We will break this sentence into shorter clauses, which will allow us to explain what we mean by "different variability".

There is an increase in the relative contribution of foraminifera from the Miocene to the Quaternary (or a decrease in the relative contribution of coccoliths), which is consistent with the findings of Si and Rosenthal (2019). In addition, we observe different fluctuations in the relative contribution of foraminifera and coccoliths within the Pliocene and Miocene, indicating that the two most important carbonate producers react differently in these two periods. Furthermore, the phase relationship between the total CaCO$_3$ AR fraction and the contribution of foraminifera changed between the Pliocene and Miocene (Figure 1).

*L210: it **changes** between; For **these two** time intervals we do not; -- this is actually confusing because then you're talking about Quaternary cold events… so are we still talking about the Pliocene and Miocene? I'm lost.*

We will apply these changes to the paragraph in the revised version of the manuscript. We understand the reviewer concern, the mention of the Quaternary into brackets was just for clarification, we will remove it.

*L214: '…proof of the preferential dissolution…' you're arguing it constitutes 'proof' based on not a single scrap of evidence or explanation! The rest of the manuscript seems carefully constructed to build arguments based on observation, so I'm not sure what the blind spot is about this one thing. But it needs fixing.*

Yes, this has also been addressed by Reviewer 2 and we must address the influence of carbonate dissolution, even if it is minor.

As indicated above, this evidence exists and we will include an appropriate section on this. We were perhaps carried away by the existence of such discussion in our previous paper (Cornuault et al., 2023) and thus felt it does not need to be reiterated here, but the comments of the referee show that it is preferrable to have the evidence presented in this paper as well.

Ultimately, we cannot avoid including a chapter in the (so far rather brief) discussion in which we discuss possible factors influencing the different carbonate contents and carbonate accumulation of the various grain size fractions. We will also discuss the possibility that the total carbonate does not consist solely of coccoliths and (planktonic) foraminifera, but can also contain pteropods, and possibly benthic foraminifera and ostracods, etc., which can influence the total carbonate and coarse carbonate >63μm. We will additionally mention that coccoliths may not be the only formers of fine carbonate <32μm – fragments of foraminifera or even pteropods can play a minor role here, as can perhaps also redeposited fine carbonate (from river or wind transport, who knows?).

*L217: carbonate calcifiers, **while remaining** similar on **a** geologic time scale*

This will be changed in the revised version of the manuscript.

*L218: 'it isn't' should be **it is not** (don't usually use contractions in formal writing)*

Sure, it will be modified in the final version of the manuscript.

*L219: CaCO3 production **that** is driving the*

We will modify it for the final version of the manuscript.

*L218-219: How does this work with the absence of a correlation?*

This sentence indeed needs an explanation: this conclusion was made by Cornuault et al. (2023) on the basis of the variance showed by the reconstructed flux of the individual components and of the total carbonate. The total carbonate flux cannot be driven by variations in the coarse fraction flux, because this varies much less than the total carbonate flux. One can see it in Figure 1. We will add an explanation as to where this information comes from.

This works because for these two time intervals, the spectral analysis are showing that the $CaCO_3$ AR bulk changes are following exactly the same orbital variability than the <63 μm $CaCO_3$ production, and not the one of the coarse fraction $CaCO_3$ production. Additionally, it is visible as well on figure 1 that most of the $CaCO_3$ AR bulk production is explained by <63 μm $CaCO_3$ production. We will reword it to make it clearer.

*L221: remove the comma after (2019)*

It will be changed in the final version of the manuscript.

*L223-224: time scale, **producing changes in their** relative contribution to the CaCO3 AR bulk**.***

We do think that the word "producing" is not reflecting exactly the point we want to make but understand the reviewer's concern, we plan to reword it as "[…] time scale, **resulting in a change of** their relative contribution to the CaCO3 AR bulk."

*L227: covary **nor are they** linked*

This will be modified in the final version of the manuscript.

*L227-228: Point 1 seems a bit repetitive? They're not linked, meaning they're not linked…*

We thank the reviewer for rising that point, it will be reworded as "[…] changing through the time without responding to the same forcing […]"

*L231: AR bulk **increases, both groups increase but not necessarily to the same degree** (e.g.*

We thank the reviewer for this comment and will modify the sentence according to it, except for the word "degree" for which we think that the word "amplitude" is more appropriate.

*L233: increase **in** eccentricity corresponds to an increase **in** the mean*

This will be changed in the final version of the manuscript.

*L235: due to **their** dependence*

This will be modified in the final version of the manuscript.

*L236: Why would a larger number of species cause them to be more eccentricity affected? This could do with elaboration, as could the evolutionary forcing of coccolithophores by eccentricity (L239) and how this affects the carbonate accumulation rate and drives carbon cycle changes. This is the part of the meat of the discussion, and these are not self-evident claims that the broad readership of this journal will necessarily be familiar with. In addition to citing the relevant papers it would greatly help the discussion if these interpretations were explained or pieced together in the main text. These could have their own paragraphs even; there's plenty of space.*

We understand that this part of the discussion requires more elaboration and we will add it, drawing on the arguments presented by Beaufort et al. (2022) from where the idea originates.

We will add a paragraph here to address the reviewer's comment and discuss the findings in more detail and in the context of literature data (and the corresponding references).

*L239: as the **result** of; **coccolithophore** evolution (or **coccolith**, but not coccoliths)*

This will be modified in the final version of the manuscript.

*L240: I would find it easier to read this section if there was a paragraph break after (Beaufort et al., 2022).*

We thank the reviewer for this suggestion and will begin the next sentence in another paragraph.

*L241: As the two **main** pelagic*

This will be changed in the final version of the manuscript.

*L244: biomineralization process… This could be more specific, and elaborated on.*

Indeed. We will remove this from the bracket and extend it into a sentence reiterating the exact argument by the author.

*L246: so if **both foraminifera and coccolith** productivity (or **coccolithophore** might be more accurate?)*

The referee is right, because productivity is the function of an organism, not its parts. The coccolithophores must have been more productive, making more coccoliths. We could also change it into "higher production of coccoliths".

This will be updated in the final version of the manuscript, but we will keep "coccoliths" as we are not measuring coccospheres but coccoliths.

*L245-246: Are we talking about the PWP? Otherwise I'm lost again, since there's purportedly no correlation…*

Yes, the sentence lacks the information on when we observe the increase. We will reword this sentence to make our point clearer.

*L248-250: Way more is needed here to flesh this out (and it could be interesting to explore!) and the English needs some work as well, and the organization (why does the last sentence happen there?).*

*L253-260: Same with this paragraph.*

We will expand on this argument as indicate above and indeed, we note that the last sentence is placed here inappropriately. It should have been deleted. This applies to the next paragraph as well.

We will expand the discussion and discuss in detail the possible factors influencing the different carbonate contents and carbonate accumulation of the various grain size fractions, as well as their effects over the time intervals considered.

*L275: trace elements**, and they sink** through the water column*

This will be modified in the final version of the manuscript.

*L277: a large change **in** the*

This will be modified in the final version of the manuscript.

*L277: 'by these two calcifiers'… by one rather than the other, you mean, presumably?*

Yes, we mean "by either of the two" and we will change the sentence accordingly.

*L282: of **>32 μm** sediment*

No, **less** than **32 μm** (**<32 μm**).

*L289-293: Surely you're not thinking of redacting it already?*

We will change the sentence to "contributed to the writing of the manuscript".

---

## Author Comment (AC2)

**Response to reviewer 2**

We thank the reviewers for their thoughtful and constructive comments, which will help us improve the quality of our manuscript. Below we respond to each point raised, and indicate when changes will be made in the manuscript. Reviewer's comments are in Italic font and our responses are plane text.

*Major Comments:*
*1. Biological vs. Sedimentological Signals:*

*The assumption that size-fraction partitioning reflects biological production differences between foraminifera and coccolithophores is plausible but deserves a more explicit discussion of potential confounding effects (e.g., selective dissolution, fragmentation, lateral transport). The authors should clarify whether assemblage data or size-specific calcification trends support the interpretation of biogenic variability.*

We deliberately chose this location because we know, as mentioned L72 to L76, that during the time intervals studied, carbonate dissolution or fragmentation, as well as sediment disturbances, are negligible and that the $CaCO_3$ AR of each class size essentially reflects the biological production differences between foraminifera and coccolithophores. We are aware that there may be very slight fragmentation or carbonate dissolution, but that does not have to be obviously recognisable and may also not necessarily influence the findings. We will add a discussion on how dissolution or winnowing "would affect" the signal if they occurred. Fragmentation is the same as dissolution. We also should add that we agree that indeed assemblage composition data and size data would help us to understand exactly which biological factors are responsible for the variability. We intend to do this in the final paper. Thus, we will take up the aspects raised by the reviewer and incorporate them into our updated discussion, as a similar comment was also made by Reviewer 1.

*2. Mechanisms of Orbital Response:*

*The manuscript convincingly shows that foraminifera and coccolithophores respond to different orbital frequencies and with differing phase relationships. However, the ecological or physiological mechanisms behind these responses (e.g., light dependence, seasonality, nutrient sensitivity) should be elaborated further. Drawing on trait-based frameworks or previous ecological modelling could help anchor these interpretations more firmly.*

This is addressed in the last point of this document.

*3. Assemblage Data:*

*The absence of plankton assemblage composition or species-specific trends is a limitation, particularly when interpreting variability in calcite flux. If such data exist from the same samples or nearby cores, even qualitative discussion would add significant value.*

We will explore this in a new paper. There are no assemblages composition for the foraminifera but we have counts and we do have a distinction of species or groups of species for the coccoliths with counts. Added to this we have morphometric measurements, but all these data with their analyses and interpretations, given their importance, will be the object of a separate manuscript and publication.

*4. Future Climate Relevance:*

*The discussion raises intriguing questions about the potential impact of future warming on pelagic carbonate production and ocean alkalinity. However, these speculations should be more tightly linked to modern observations or modelling results and framed more cautiously, acknowledging the complexity of projecting ecological responses from past analogues.*

This is addressed in the last point of this document.

***Specific Suggestions:***

- *Justify the use of the 63 µm cutoff for foraminifera in greater detail, especially in light of evidence for small but non-negligible contributions from smaller foraminifera (Langer, 2008; Schiebel, 2002 may be of relevance.)*

We thank the reviewer for this helpful suggestion. We plan to add references on the topic of the use of the 63 µm cutoff for foraminifera and the possible associated bias in the revised material and methods L90 to L93.

- *Include uncertainty estimates (e.g., error propagation in $CaCO_3$ accumulation rate calculations) and statistical support (e.g., $r^2$ and p-values in regression plots).*

Regarding r2 and p values in regression plots, they are already included in figure 2.

- *Improve figure captions by clearly stating the methods used and adding key interpretation notes where appropriate.*

This will be done in the revised version of the manuscript, both in the main text and in the supplements.

- *Clarify the limitations of the SYRACO method more explicitly in the main text, particularly regarding the exclusion of aggregates or poorly preserved specimens.*

A sentence will be added in the revised material and methods, between L118 and L119, stating that the SYRACO device does not recognise aggregates or poorly preserved specimens, resulting in their exclusion from the final data. Nevertheless, the specimens are well preserved at the chosen location and the used protocol avoids aggregates, therefore, these scenarios remain rare.

- *Summarise key results from the supplementary wavelet analyses in a main-text synthesis figure (e.g., a matrix showing orbital periodicity detection across taxa and time intervals).*

This will be added in the revised manuscript.

***Minor Points:***
*- Check grammar and conciseness in the discussion section. Examples:*

- *L225-230: very long sentence – try to break it down into 2 sentences at least*

We thank the reviewer for pointing this out, we will rewrite it for the revised version of the manuscript as "According to the observations of the wavelet transform and $CaCO_3$ AR records vs time (Figures S1-S3), the foraminifera and the coccoliths are not responding to the same orbital parameter and not in the same way. The changes of $CaCO_3$ AR of the two groups do not seem to covary nor are they linked

by any correlation, meaning that they are changing through the time without responding to the same forcing. This could also explain the different frequencies observed in the bulk $CaCO_3$ AR record (Cornuault et al., 2023): one frequency tracking coarse (foraminifera) and one frequency tracking small (coccolithophores) fraction."

- *L223: "causing" instead of "making"?*

We will reword it for the revised version of the manuscript as "[…] time scale, resulting in a change of their relative contribution to the $CaCO_3$ AR bulk."

- *L227: "or be linked" instead of "or being linked"*

This will be modified as mentioned above.

- *L238: Define who/what "Their" is – coccolithophores? And how do we know their morphology changes with eccentricity? Provide a reference*

"Their" is indeed coccolithophores, it will be changed in the final version of the manuscript as "The coccolithophores morphological diversity […]". Regarding the reference, it is already there at the end of the sentence, L40: "(Beaufort et al., 2022)".

- *L241: Be more precise than "changed a lot"*

We thank the reviewer for highlighting this lack of precision, we will rewrite the sentence as: "We observe important differences of the orbitally forced variability within the different time intervals and different responses between the two groups."

- *L249: Expand on "This would reduce the ocean's carbon sink capacity", provide references, no need to capitalise ocean in this instance*

We thank the reviewer for this helpful suggestion. We will add a more detailed explanation of how we arrived at this conclusion by rewriting the lines L248, L249: "Can we expect a decrease in the role of the coccoliths in the surface ocean anthropogenic carbon absorption in the future with warmer climates? This would reduce the Ocean's carbon sink capacity (Ziveri et al., 2023)."

- *L250-253: Rewrite for clarity and define "that much"*

We will rewrite this portion for clarity and reword as follow: "Our results are in line with those obtained by Schwab et al. (2013) and Stolz and Baumann (2010) who find a high coccolith productivity during the MIS 5e, even if during this interval, the coarse fraction increased with such amplitude that there is a relative increase of the foraminifera fraction in the sediment."

- *L254: Use correct symbol for delta, e.g. $\delta^{18}O$*

This will be modified in the revised version of the manuscript.

- *L276 to end: Expand on why this is an issue*

We thank the reviewer for pointing this out, we will add a more extended explanation of why this is an issue by rewriting the end of the manuscript and breaking the sentences into three. First, we explain

the effect of differential susceptibility, then, we explain the effect of different trace elements, finally, we explain the effect of different ballasting.

*- Define all abbreviations (e.g., CBT, ETP) upon first use in figures and text.*

This will be checked and added in the revised version of the manuscript.

*- Include references to relevant recent studies on coccolithophore or foraminiferal responses to modern climate change, where discussing implications.*

We thank the reviewer for this helpful suggestion. We will add a paragraph with recent references in coccolithophore and foraminiferal responses to modern climate change, and their possible path in the future. This will be added in the discussion to compare my data with and give context to the possible projections of our findings.

Modern climate change includes changes in $pCO_2$ (and $CO_2$ dissolved in ocean waters, resulting in changes of alkalinity), temperature (Globally, SSTs have risen by an average of 0.97°C (confidence interval: [0.77°C - 1.09°C]) between 1850-1900 and 2014-2023 (Forster et al., 2024)) and ocean pH (Intergovernmental Panel On Climate Change, 2021) and chemical conditions for calcification have become less favorable over the past 40 years (Bates and Johnson, 2023).

Regarding the coccolithophores, recent studies found a decreasing calcium carbonate production relative to growth with increasing p $CO_2$ (and decreasing $CO_3^{2-}$ concentrations) in most areas (except for one E. huxleyi morphotype), with species-specific response. The PIC/POC ratio of coccolithophores decreases with increasing $CO_2$ (Gafar et al., 2018) and this modifies (marine) carbon cycle, regional carbon export and ocean-atmosphere $CO_2$ exchange feedbacks (Beaufort et al., 2011 -cultures-; Krumhardt et al., 2019 -models-). Coccolithophores become more abundant but less calcified as $CO_2$ increases with a tipping point in global calcification (changing from increasing to decreasing calcification relative to preindustrial) at approximately $\sim$600 $\mu$atm $CO_2$ (Krumhardt et al., 2019). With increasing temperature and surface ocean p $CO_2$, their calcification and growth rate decreases in most low and mid latitude regions, with possible increases in both of these responses in most high latitude regions." (Krumhardt et al., 2017) and some studies found coccolithophores blooms to be limited by temperature increase (Oliver et al., 2024). Furthermore, the time scale matters, as phytoplankton can adapt to temperature increases as long as they occur over the time scale of a century but when rapid and extreme events of temperature change are considered, the phytoplankton adaptive capacity breaks down and primary productivity plummets (Sauterey et al., 2023).

Coccolithophore dynamics are expected to be driven by ocean warming and stratification, as evidenced by their poleward shift in response to rising sea surface temperatures (D'Amario et al., 2020; Meyer and Riebesell, 2015; Hutchins and Tagliabue, 2024), and the coccolithophore communities are described as varying with depth and latitude (Han et al., 2025). In recent years, high latitude regions are being potential refuges, with both coccolithophore and foraminifera expanding towards poles (Chaabane et al., 2024; Winter et al., 2014; Ying et al., 2024). The long-term stability of high latitudes is highly uncertain under continued warming, ocean acidification, and ecosystem restructuring (Guinaldo and Neukermans, 2025) and it is unknown if they will sustain prolonged darkness at higher latitudes, or water freshening trends in the high-latitude North Atlantic and Southern Oceans (Cheng et al., 2020).

Regarding the foraminifera, it has been shown that their response to ocean acidification and warming is species-specific and complex as well. Models show species-specific responses of planktonic

foraminifera shell weight changes with increasing p $CO_2$, associated with drivers including (but not limited to) the carbonate system, which are likely different between ocean basins (Barrett et al., 2025). A weight reduction of some species has been observed between pre-industrial and post-industrial Holocene and recent data, with G. truncatulinoides experiencing the largest weight loss (32 %–40 %) followed by G. bulloides (18 % - 24 %) and N. incompta (9 %–18 %), evidence of a decrease in planktonic foraminifera calcification (Béjard et al., 2023). A 37.9 % decrease in total foraminiferal flux for the years 2014-2021 relative to the 1990s, has been observed accompanied by a 21.9 % overall reduction in calcium carbonate flux and a decrease in the relative abundance of subtropical species. The extremely rapid responses of foraminifera suggest that climate change is already having a meaningful impact on coastal carbon cycling and the observed decrease in particulate inorganic carbon (PIC) flux relative to particulate organic carbon (POC) flux may facilitate increased oceanic uptake of atmospheric $CO_2$ (Havard et al., 2024). They are projected to reduce their global carbon biomass between 8% (RCP6) and 11% (RCP8.5) by 2050 and between 14% and 18% by 2100 as a response to ocean warming and associated changes in primary production and ecological dynamics by 2100 relative to 1900–1950. That decline can slow down the ocean carbonate pump and create short-term positive feedback on rising atmospheric p $CO_2$ (Grigoratou et al., 2022; Ying et al., 2024).

The calcification seems to be depending on seawater density with a buoyancy regulatory function, and foraminifera bulk shell densities may serve as a seawater density proxy with calcification responding to the anthropogenically driven reductions in ocean density, with potential consequences for the carbon cycle (Zarkogiannis et al., 2025). Foraminifera acclimatization capacities are limited and insufficient to track warming rates and migration will not be enough to ensure survival (Chaabane et al., 2024). Certain species or strains may acclimate to future ocean pH and temperature (Rigual-Hernández et al., 2020; WestgÅrd et al., 2023), whereas others may perish or expand to new ecological niches where conditions are more suitable (Shemi et al., 2025), modifying the carbon export by the carbonate counter pump (Neukermans et al., 2023).

Satellite data are to be taken carefully, as it has been shown that high reflectance signal may relate to the presence of small biogenic opal particles or other unknown highly reflective particles, and be the reason of high satellite-derived PIC concentrations that are not apparent in the coccolith-based PIC data (Saavedra-Pellitero et al., 2025). A strong overestimation of PIC by sensor (and even more so by satellites) in surface waters colder than 1°C has also been found by Li et al. (2025). Ziveri et al. (2023) observe a decoupling of $CaCO_3$ production and export, with pelagic $CaCO_3$ production being higher than the sinking flux of $CaCO_3$ at 150 and 200 m, implying the remineralisation of large portion of pelagic calcium carbonate within the photic zone, explaining the apparent discrepancy between previous estimates of $CaCO_3$ production derived from satellite observations/biogeochemical modelling versus estimates from shallow sediment traps. The relative abundance of foraminifera to coccolithophores has been highlighted as an important factor that could lead to large changes in the amount of $CaCO_3$ exported from the surface ocean and thus the cycle of alkalinity (Ziveri et al., 2023).

The current study is and its conclusions regarding the possible response of pelagic calcifiers in the actual context of climate warming are coherent with previously published findings.

*Bibliography*

Barrett, R., De Vries, J., and Schmidt, D. N.: What controls planktic foraminiferal calcification?, Biogeosciences, 22, 791–807, https://doi.org/10.5194/bg-22-791-2025, 2025.

Bates, N. R. and Johnson, R. J.: Forty years of ocean acidification observations (1983–2023) in the Sargasso Sea at the Bermuda Atlantic Time-series Study site, Front. Mar. Sci., 10, 1289931, https://doi.org/10.3389/fmars.2023.1289931, 2023.

Beaufort, L., Probert, I., de Garidel-Thoron, T., Bendif, E. M., Ruiz-Pino, D., Metzl, N., Goyet, C., Buchet, N., Coupel, P., Grelaud, M., Rost, B., Rickaby, R. E. M., and de Vargas, C.: Sensitivity of coccolithophores to carbonate chemistry and ocean acidification, Nature, 476, 80–83, https://doi.org/10.1038/nature10295, 2011.

Béjard, T. M., Rigual-Hernández, A. S., Flores, J. A., Tarruella, J. P., Durrieu De Madron, X., Cacho, I., Haghipour, N., Hunter, A., and Sierro, F. J.: Calcification response of planktic foraminifera to environmental change in the western Mediterranean Sea during the industrial era, Biogeosciences, 20, 1505–1528, https://doi.org/10.5194/bg-20-1505-2023, 2023.

Chaabane, S., De Garidel-Thoron, T., Meilland, J., Sulpis, O., Chalk, T. B., Brummer, G.-J. A., Mortyn, P. G., Giraud, X., Howa, H., Casajus, N., Kuroyanagi, A., Beaugrand, G., and Schiebel, R.: Migrating is not enough for modern planktonic foraminifera in a changing ocean, Nature, 636, 390–396, https://doi.org/10.1038/s41586-024-08191-5, 2024.

Cheng, L., Trenberth, K. E., Gruber, N., Abraham, J. P., Fasullo, J. T., Li, G., Mann, M. E., Zhao, X., and Zhu, J.: Improved Estimates of Changes in Upper Ocean Salinity and the Hydrological Cycle, Journal of Climate, 33, 10357–10381, https://doi.org/10.1175/JCLI-D-20-0366.1, 2020.

D'Amario, B., Pérez, C., Grelaud, M., Pitta, P., Krasakopoulou, E., and Ziveri, P.: Coccolithophore community response to ocean acidification and warming in the Eastern Mediterranean Sea: results from a mesocosm experiment, Sci Rep, 10, 12637, https://doi.org/10.1038/s41598-020-69519-5, 2020.

Forster, P. M., Smith, C., Walsh, T., Lamb, W. F., Lamboll, R., Hall, B., Hauser, M., Ribes, A., Rosen, D., Gillett, N. P., Palmer, M. D., Rogelj, J., Von Schuckmann, K., Trewin, B., Allen, M., Andrew, R., Betts, R. A., Borger, A., Boyer, T., Broersma, J. A., Buontempo, C., Burgess, S., Cagnazzo, C., Cheng, L., Friedlingstein, P., Gettelman, A., Gütschow, J., Ishii, M., Jenkins, S., Lan, X., Morice, C., Mühle, J., Kadow, C., Kennedy, J., Killick, R. E., Krummel, P. B., Minx, J. C., Myhre, G., Naik, V., Peters, G. P., Pirani, A., Pongratz, J., Schleussner, C.-F., Seneviratne, S. I., Szopa, S., Thorne, P., Kovilakam, M. V. M., Majamäki, E., Jalkanen, J.-P., Van Marle, M., Hoesly, R. M., Rohde, R., Schumacher, D., Van Der Werf, G., Vose, R., Zickfeld, K., Zhang, X., Masson-Delmotte, V., and Zhai, P.: Indicators of Global Climate Change 2023: annual update of key indicators of the state of the climate system and human influence, Earth Syst. Sci. Data, 16, 2625–2658, https://doi.org/10.5194/essd-16-2625-2024, 2024.

Gafar, N. A., Eyre, B. D., and Schulz, K. G.: A Conceptual Model for Projecting Coccolithophorid Growth, Calcification and Photosynthetic Carbon Fixation Rates in Response to Global Ocean Change, Front. Mar. Sci., 4, 433, https://doi.org/10.3389/fmars.2017.00433, 2018.

Grigoratou, M., Monteiro, F. M., Wilson, J. D., Ridgwell, A., and Schmidt, D. N.: Exploring the impact of climate change on the global distribution of non-spinose planktonic foraminifera using a trait-based ecosystem model, Global Change Biology, 28, 1063–1076, https://doi.org/10.1111/gcb.15964, 2022.

Han, Y., Steiner, Z., Cao, Z., Fan, D., Chen, J., Yu, J., and Dai, M.: Coccolithophore abundance and production and their impacts on particulate inorganic carbon cycling in the western North Pacific, Biogeosciences, 22, 3681–3697, https://doi.org/10.5194/bg-22-3681-2025, 2025.

Havard, E., Cherry, K., Benitez-Nelson, C., Tappa, E., and Davis, C. V.: Decreasing foraminiferal flux in response to ongoing climate change in the Santa Barbara Basin, California, https://doi.org/10.5194/egusphere-2024-3374, 5 December 2024.

Hutchins, D. A. and Tagliabue, A.: Feedbacks between phytoplankton and nutrient cycles in a warming ocean, Nat. Geosci., 17, 495–502, https://doi.org/10.1038/s41561-024-01454-w, 2024.

Intergovernmental Panel On Climate Change: Climate Change 2021 – The Physical Science Basis: Working Group I Contribution to the Sixth Assessment Report of the Intergovernmental Panel on Climate Change, 1st ed., Cambridge University Press, https://doi.org/10.1017/9781009157896, 2021.

Krumhardt, K. M., Lovenduski, N. S., Iglesias-Rodriguez, M. D., and Kleypas, J. A.: Coccolithophore growth and calcification in a changing ocean, Progress in Oceanography, 159, 276–295, https://doi.org/10.1016/j.pocean.2017.10.007, 2017.

Krumhardt, K. M., Lovenduski, N. S., Long, M. C., Levy, M., Lindsay, K., Moore, J. K., and Nissen, C.: Coccolithophore Growth and Calcification in an Acidified Ocean: Insights from Community Earth System Model Simulations, J. Adv. Model. Earth Syst., 11, 1418–1437, https://doi.org/10.1029/2018MS001483, 2019.

Li, Y., Bishop, J. K. B., Lam, P. J., and Ohnemus, D. C.: Analysis of Satellite and In Situ Optical Proxies for PIC and POC During GEOTRACES GP15 and GP17-OCE Transects From the Subarctic North Pacific to the Southern Ocean, Earth and Space Science, 12, e2024EA004070, https://doi.org/10.1029/2024EA004070, 2025.

Meyer, J. and Riebesell, U.: Reviews and Syntheses: Responses of coccolithophores to ocean acidification: a meta-analysis, Biogeosciences, 12, 1671–1682, https://doi.org/10.5194/bg-12-1671-2015, 2015.

Neukermans, G., Bach, L. T., Butterley, A., Sun, Q., Claustre, H., and Fournier, G. R.: Quantitative and mechanistic understanding of the open ocean carbonate pump - perspectives for remote sensing and autonomous in situ observation, Earth-Science Reviews, 239, 104359, https://doi.org/10.1016/j.earscirev.2023.104359, 2023.

Oliver, H., Krumhardt, K. M., McGillicuddy, D. J., Mitchell, C., and Balch, W. M.: Mechanisms Regulating Coccolithophore Dynamics in the Great Calcite Belt in the Southern Ocean in the Community Earth System Model, JGR Oceans, 129, e2024JC021371, https://doi.org/10.1029/2024JC021371, 2024.

Rigual-Hernández, A. S., Trull, T. W., Flores, J. A., Nodder, S. D., Eriksen, R., Davies, D. M., Hallegraeff, G. M., Sierro, F. J., Patil, S. M., Cortina, A., Ballegeer, A. M., Northcote, L. C., Abrantes, F., and Rufino, M. M.: Full annual monitoring of Subantarctic *Emiliania huxleyi* populations reveals highly calcified morphotypes in high-$CO_2$ winter conditions, Sci Rep, 10, 2594, https://doi.org/10.1038/s41598-020-59375-8, 2020.

Saavedra-Pellitero, M., Baumann, K.-H., Bachiller-Jareno, N., Lovell, H., Vollmar, N. M., and Malinverno, E.: Mismatch between coccolithophore-based estimates of particulate inorganic carbon (PIC) concentration and satellite-derived PIC concentration in the Pacific Southern Ocean, Biogeosciences, 22, 3143–3164, https://doi.org/10.5194/bg-22-3143-2025, 2025.

Sauterey, B., Gland, G. L., Cermeño, P., Aumont, O., Lévy, M., and Vallina, S. M.: Phytoplankton adaptive resilience to climate change collapses in case of extreme events – A modeling study, Ecological Modelling, 483, 110437, https://doi.org/10.1016/j.ecolmodel.2023.110437, 2023.

Shemi, A., Gal, A., and Vardi, A.: Uncertain fate of pelagic calcifying protists: a cellular perspective on a changing ocean, The ISME Journal, wraf007, https://doi.org/10.1093/ismejo/wraf007, 2025.

WestgÅrd, A., Ezat, M. M., Chalk, T. B., Chierici, M., Foster, G. L., and Meilland, J.: Large-scale culturing of *Neogloboquadrina pachyderma* , its growth in, and tolerance of, variable environmental conditions, Journal of Plankton Research, 45, 732–745, https://doi.org/10.1093/plankt/fbad034, 2023.

Winter, A., Henderiks, J., Beaufort, L., Rickaby, R. E. M., and Brown, C. W.: Poleward expansion of the coccolithophore *Emiliania huxleyi*, Journal of Plankton Research, 36, 316–325, https://doi.org/10.1093/plankt/fbt110, 2014.

Ying, R., Monteiro, F. M., Wilson, J. D., Ödalen, M., and Schmidt, D. N.: Past foraminiferal acclimatization capacity is limited during future warming, Nature, 636, 385–389, https://doi.org/10.1038/s41586-024-08029-0, 2024.

Zarkogiannis, S. D., Rae, J. W. B., Shipley, B. R., and Mortyn, P. G.: Planktonic foraminifera regulate calcification according to ocean density, Commun Earth Environ, 6, 605, https://doi.org/10.1038/s43247-025-02558-w, 2025.

Ziveri, P., Gray, W. R., Anglada-Ortiz, G., Manno, C., Grelaud, M., Incarbona, A., Rae, J. W. B., Subhas, A. V., Pallacks, S., White, A., Adkins, J. F., and Berelson, W.: Pelagic calcium carbonate production and shallow dissolution in the North Pacific Ocean, Nat Commun, 14, 805, https://doi.org/10.1038/s41467-023-36177-w, 2023.